# MixCon: Adjusting the Separability of Data Representations for Harder Data Recovery

## Abstract

To address the issue that deep neural networks (DNNs) are vulnerable to model inversion attacks, we design an objective function, which adjusts the separability of the hidden data representations, as a way to control the trade-off between data utility and vulnerability to inversion attacks. Our method is motivated by the theoretical insights of data separability in neural networking training and results on the hardness of model inversion. Empirically, by adjusting the separability of data representation, we show that there exist sweet-spots for data separability such that it is difficult to recover data during inference while maintaining data utility.

## 1 Introduction

Over the past decade, deep neural networks have shown superior performances in various domains, such as visual recognition, natural language processing, robotics, and healthcare. However, recent studies have demonstrated that machine learning models are vulnerable in terms of leaking private data He et al. (2019); Zhu et al. (2019); Zhang et al. (2020b). Hence, preventing private data from being recovered by malicious attackers has become an important research direction in deep learning research.

Distributed machine learning Shokri & Shmatikov (2015); Kairouz et al. (2019) has emerged as an attractive setting to mitigate privacy leakage without requiring clients to share raw data. In the case of an edge-cloud distributed learning scenario, most layers are commonly offloaded to the cloud, while the edge device computes only a small number of convolutional layers for feature extraction, due to power and resource constraints Kang et al. (2017). For example, service provider trains and splits a neural network at a "cut layer," then deploys the rest of the layers to clients Vepakomma et al. (2018). Clients encode their dataset using those layers, then send the data representations back to cloud server using the rest of layers for inference Teerapittayanon et al. (2017); Ko et al. (2018); Vepakomma et al. (2018). This gives an untrusted cloud provider or a malicious participant a chance to steal sensitive inference data from the output of "cut layer" on the edge device side, i.e. inverting data from their outputs Fredrikson et al. (2015); Zhang et al. (2020b); He et al. (2019).

In the above distributed learning setup, we investigate how to design a hard-to-invert *data representation* function (or *hidden data representation* function), which is defined as the output of the neural network's intermediate layer. We focus on defending data recovery during inference. The goal is to hide sensitive information and to protect data representations from being used to reconstruct the original data while ensuring that the resulted data representations are still informative enough for decision making. We use the model inversion attack that reconstructs individual data He et al. (2019); Zhang et al. (2020b) to evaluate defense performance and model accuracy to evaluate data utility. The core question here is how to achieve the goal, especially protecting individual data from being recovered.

We propose *data separability*, also known as the minimum (relative) distance between (the representation of) two data points, as a new criterion to investigate and understand the trade-off between data utility and hardness of data recovery. Recent theoretical studies show that if data points are separable in the hidden embedding space of a DNN model, it is helpful for the model to achieve good classification accuracy Allen-Zhu et al. (2019b). However, larger separability is also easier to recover inputs. Conversely, if the embeddings are non-separable or sometimes overlap with one another, it is challenging to recover inputs. Nevertheless, the model may not be able to learn to achieve good performance. Two main questions arise. First, is there an effective way to adjust the separability of data representations? Second, are there "sweet spots" that make the data representations difficult for inversion attacks while achieving good accuracy.

This paper aims to answer these two questions by learning a feature extractor that can adjust the separability of data representations embedded by a few neural network layers. Specifically, we propose to add a self-supervised learning-based novel regularization term to the standard loss function during training. We conduct experiments on both synthetic and benchmark datasets to demonstrate that with specific parameters, such a learned neural network is indeed difficult to recover input data while maintaining data utility.

Our contributions can be summarized as:

- To the best of our knowledge, this is the first proposal to investigate the trade-off between data utility and data recoverability from the angle of data representation separability;
- We propose a simple yet effective loss term, Consistency Loss – MixCon for adjusting data separability;
- We provide the theoretical-guided insights of our method, including a new exponential lower bound on approximately solving the network inversion problem, based on the Exponential Time Hypothesis (ETH); and
- We report experimental results comparing accuracy and data inversion results with/without incorporating MixCon. We show MixCon with suitable parameters makes data recovery difficult while preserving high data utility.

The rest of the paper is organized as follow. We formalize our problem in Section 2. In Section 3, we present our theoretical insight and introduce the consistency loss. We demonstrate the experiment results in Section 4. We defer the technical proof and experiment details to Appendix.

## 2 PRELIMINARY

**Distributed learning framework.** We consider a distributed learning framework, in which *users* and *servers* collaboratively perform inferences Teerapittayanon et al. (2017); Ko et al. (2018); Kang et al. (2017). We have the following assumptions: 1) Datasets are stored at the user sides. During inference, no raw data are ever shared among users and servers; 2) Users and servers use a split model Vepakomma et al. (2018) where users encode their data using our proposed mechanism to extract data representations at a cut layer of a trained DNN. Servers take encoded data representations as inputs and compute outputs using the layers after the cut layer in the distributed learning setting; 3) DNN used in the distributed learning setting can be regularized by our loss function (defined later).

**Threat model.** We consider the attack model with access to the shared hidden data representations during the client-cloud communication process. The attacker aims to recover user data (i.e., pixel-wise recovery for images in vision task). To quantify the upper bound of privacy leakage under this threat model, we allow the attacker to have more power in our evaluation. In addition to having access to extracted features, we allow the attacker to see all network parameters of the trained model.

**Problem formulation.** Formally, let $h : \mathbb{R}^d \to \mathbb{R}^m$ denote the local feature extractor function, which maps an input data $x \in \mathbb{R}^d$ to its feature representation $h(x) \in \mathbb{R}^m$. The local feature extractor is a shallow neural network in our setting. The deep neural network on the server side is denoted as $g : \mathbb{R}^m \mapsto \mathbb{R}^C$, which performs classification tasks and maps the feature representation to one of $C$ target classes. The overall neural network $f : \mathbb{R}^d \mapsto \mathbb{R}^C$, and it can be written as $f = g \circ h$.

Our overall objectives are:
- Learn the feature representation mechanism (i.e. $h$ function) that safeguards information from unsolicited disclosure.
- Jointly learn the classification function $g$, and the feature extraction function $h$ to ensure the information extracted is useful for high-performance downstream tasks.

## 3 CONSISTENCY LOSS FOR ADJUSTING DATA SEPARABILITY

To address the issue of data recovery from hidden layer output, we propose a novel consistency loss in neural network training, as shown in Figure 1. Consistency loss is applied to the feature extractor

$h$ to encourage encoding closed but separable representations for the data of different classes. Thus, the feature extractor $h$ can help protect original data from being inverted by an attacker during inference while achieving desirable accuracy.

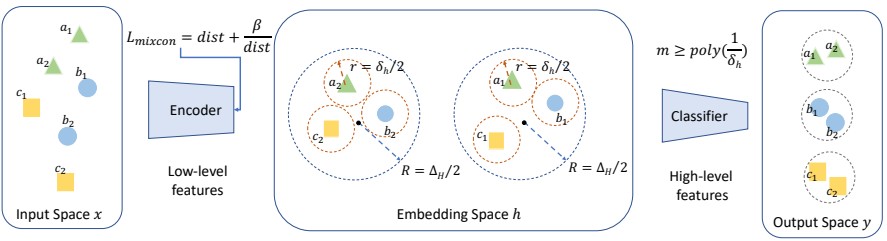

Figure 1: Schematic diagram of our data representation encoding scheme in deep learning pipeline. We show a simple toy example of classifying data points of triangles, squares, and circles. In embedding space (the middle block), data representations from different classes are constrained to a small ball with diameter $\Delta_H$, while they are separate from each other at least with distance $\delta_h$.

### 3.1 DATA SEPARATION AS A GUIDING TOOL

Our intuition is to adjust the information in the data representations to a minimum such that downstream classification tasks can achieve good accuracy but not enough for data recovery through model inversion attacks He et al. (2019). The question is, what is the right measure on the amount of information for successful classification and data security? We propose to use data separability as the measure. This intuition is motivated by the theoretical results of deep learning. In particular,

- Over-parameterized deep learning theory — the well separated data requires a narrower network to train,
- In-approximation theory — the worse separability of the data, the harder of the inversion problem.

**Definition 3.1** (Data separability). *Let $\delta_h$ denote the separability of hidden layer over all pairwise inputs $x_1, x_2, \cdots, x_n \in \mathbb{R}^d$, i.e.,*

$$\delta_h := \min_{i \neq j \in [n]} \|h(x_i) - h(x_j)\|_2. \quad \text{(controlling accuracy)}$$

*Let $S$ denote a set of pairs that supposed to be close in hidden layer. Let $\Delta_H$ denote the maximum distance with respect to that set $S$*

$$\Delta_H := \max_{(i,j) \in S} \|h(x_i) - h(x_j)\|_2. \quad \text{(controlling invertibility)}$$

Intuitively, we expect the upper bound on data separability ($\Delta_H$) relates to the invertibility and the lower bound on data separability ($\delta_h$) relates to the accuracy.

**Lower bound on data separability implies better accuracy** Recent line of deep learning theory Allen-Zhu et al. (2019b) indicates that data separability is perhaps the only matter fact for learnability (at least for overparameterized neural network), leading into the following results.

**Theorem 3.2.** *Allen-Zhu et al. (2019b) Suppose the training data points are separable, i.e., $\delta_h > 0$. If the width of a $L$-layer neural network with ReLU gates satisfies $m \geq \text{poly}(n, d, L, 1/\delta_h)$, initializing from a random weight matrix $W$, (stochastic) gradient descent algorithm can find the global minimum of neural network function $f$.*

Essentially, the above theorem indicates that we can (provably) find global minimum of a neural network given well separated data, and better separable data points requires narrower neural network and less running time.

**Upper bound on data separability implies hardness of inversion.** When all data representation is close to each other, i.e. $\Delta_H$ is sufficiently small, we expect the inversion problem is hard. We

support this intuition by proving that the neural network inversion problem is hard to *approximate* within some constant factor when assuming NP$\neq$RP.[1]

Existing work Lei et al. (2019) indicates that the decision version of the neural network inversion problem is NP-hard. However, this is insufficient since it is usually easy to find an approximate solution, which could leak much information on the original data. It is an open question whether the approximation version is also challenging. We strengthen the hardness result and show that by assuming NP$\neq$RP, it is hard to recovery an input that approximates the hidden layer representation. Our hardness result implies that given hidden representations are close to each other, no polynomial time can distinguish their input. Therefore, it is impossible to recover the real input data in polynomial time.

**Theorem 3.3** (Informal). *Assume* NP$\neq$RP*, there is no polynomial time algorithm that is able to give a constant approximation to thee neural network inversion problem.*

The above result only rules out the polynomial running time recovery algorithm but leaves out the possibility of a subexponential time algorithm. To further strengthen the result, we assume the well-known Exponential Time Hypothesis (ETH), which is widely accepted in the computation complexity community.

**Hypothesis 3.4** (Exponential Time Hypothesis (ETH) Impagliazzo et al. (1998)). *There is a $\delta > 0$ such that the* 3SAT *problem cannot be solved in $O(2^{\delta n})$ time.*

Assuming ETH, we derive an exponential lower bound on approximately recovering the input.

**Corollary 3.5** (Informal). *Assume* ETH*, there is no $2^{o(n^{1-o(1)})}$ time algorithm that is able to give a constant approximation to neural network inversion problem.*

### 3.2 Consistency loss — MixCon

Follow the above intuitions, we propose a novel loss term MixCon loss — $\mathcal{L}_{\mathrm{mixcon}}$ — to incorporate in training. MixCon adjusts data separability by forcing the consistency of hidden data representations from different classes. This additional loss term balances the data separability, punishing feature representations that are too far or too close to each other. Noting that we choose to mix data from different classes instead of the data within a class, in order to bring more confusion in embedding space and potentially hiding data label information[2].

**MixCon loss $\mathcal{L}_{\mathrm{mixcon}}$:** We add consistency penalties to force the data representation of $i$-th data in different classes to be similar, while without any overlapping for any two data points.

$$\mathcal{L}_{\mathrm{mixcon}} := \frac{1}{p}\frac{1}{|\mathcal{C}|\cdot(|\mathcal{C}|-1)}\sum_{i=1}^{p}\sum_{c_1\in\mathcal{C}}\sum_{c_2\in\mathcal{C}}(\mathrm{dist}(i,c_1,c_2)+\beta/\mathrm{dist}(i,c_1,c_2)). \tag{1}$$

A practical choice for the pairwise distance is $\mathrm{dist}(i,c_1,c_2)=\|h(x_{i,c_1})-h(x_{i,c_2})\|_2^2$ [3], where $x_{i,c}$ is the $i$-th input data point in class $c$, $p := \min_{c\in\mathcal{C}}|c|$, and $\beta > 0$ balances the data separability. Note that the order $i$ is not fixed for a data point due to random shuffling in the regular training process. Thus Eq. 1 can nearly perform as all-pair comparisons in training with data shuffling. The first term punishes large distance while the second term enforces sufficient data separability. In general, we could replace $(\mathrm{dist}+\beta/\mathrm{dist})$ by convex functions with asymptote shape on non-negative domain, that is, function with value reaches infinity on both ends of $[0,\infty)$.

We consider the classification loss

$$\mathcal{L}_{\mathrm{class}} := -\sum_{i=1}^{N}\sum_{c=1}^{C}y_{i,c}\cdot\log(\widehat{y}_{i,c}) \quad \text{(cross entropy)} \tag{2}$$

---

[1]The class RP consists of all languages $L$ that have a polynomial-time randomized algorithm $A$ with the following behavior: If $x \notin L$, then A always rejects $x$ (with probability 1). If $x \in L$, then A accepts $x$ in L with probability at least $1/2$.

[2]We show the comparison in Appendix D.1.

[3]In practice, we normalize $\|h(x)\|_2$ to 1. To avoid division by zero, we can use a positive small $\epsilon$ ($\ll 1$) and threshold distance to the range of $[\epsilon, 1/\epsilon]$.

where $y_i \in \mathbb{R}^C$ is the one-hot representation of true label and $\widehat{y}_i = f(x_i) \in \mathbb{R}^C$ is the prediction score of data $i \in \{1, \dots, N\}$. The final objective function is $\mathcal{L} := \mathcal{L}_{\text{class}} + \lambda \cdot \mathcal{L}_{\text{mixcon}}$. We simultaneously train $h$ and $g$, where $\lambda$ and $\beta$ are tunable hyper-parameters associated with consistency loss regularization to adjust separability. We discuss the effect of $\lambda$ and $\beta$ in experiments (Section 4).

## 4 EXPERIMENTAL RESULTS

Our hypothesis in this work is MixCon loss can adjust the separability of data hidden representations. A moderate reduction of representation separability can still keep data utility while making it harder to recover from its representation. To show this hypothesis hold, we conduct two experiments. Synthetic experiments provide a straightforward illustration for the relationship among data separability, data utility and reversibility. Benchmark experiments are used for image data classification and recovery evaluation.

### 4.1 DATA RECOVERY MODEL

To empirically evaluate the quality of inversion, we formally define the white-box data recovery (inversion) model He et al. (2019) used in our experiments. The model aims to solve an optimization problem in the input space. Given a representation $z = h(x)$ of a testing data point, and a public function $h$ (the trained network that generates data representations), the inversion model tries to find the original input $x$:

$$x^* = \arg\min_s \mathcal{L}(h(s), z) + \alpha \cdot \mathcal{R}(s) \tag{3}$$

where $\mathcal{L}$ is the loss function that measures the similarity between $h(s)$ and $z$, and $\mathcal{R}$ is the regularization term. We specify $\mathcal{L}$ and $\mathcal{R}$ used in each experiment later. We solve Eq. (3) by iterative gradient descent.

### 4.2 EXPERIMENTS WITH SYNTHETIC DATA

To allow precise manipulation and straightforward visualization for data separability, our experiments use generated synthetic data with a 4-layer fully-connected network, such that we can control the dimensionality. In this section, we want to answer the following questions:

Q1 What is the impact of having $\beta$ in Eq.(1) to bound the smallest data pairwise distance?

Q2 Is feature encoded with MixCon mechanism harder to invert?

**Network, data generation and training.** We defined the network as

$$y = q(\text{softmax}(f(x))), \quad f(x) = W_4 \cdot \sigma(W_3 \cdot \sigma(W_2 \cdot (\sigma(W_1 x + b_1)) + b_2) + b_3) + b_4$$

$x \in \mathbb{R}^{10}$, $W_1 \in \mathbb{R}^{500 \times 10}$, $W_2 \in \mathbb{R}^{2 \times 500}$, $W_3 \in \mathbb{R}^{100 \times 2}$, $W_4 \in \mathbb{R}^{2 \times 100}$, $b_1 \in \mathbb{R}^{500}$, $b_2 \in \mathbb{R}^2$, $b_3 \in \mathbb{R}^{100}$, $b_4 \in \mathbb{R}^2$. For a vector $z$, we use $q(z)$ to denote the index $i$ such that $|z_i| > |z_j|$, $\forall j \neq i$. We initialize each entry of $W_k$ and $b_k$ from $\mathcal{N}(u_k, 1)$, where $u_k \sim \mathcal{N}(0, \alpha)$ and $k \in \{1, 2, 3, 4\}$.

We generate synthetic samples $(x, y)$ from two multivariate normal distribution. Positive data are sampled from $\mathcal{N}(0, I)$, and negative data are sampled from $\mathcal{N}(-1, I)$, ending up with 800 training samples and 200 testing samples, where the covariance matrix $I$ is an identity diagonal matrix. $\mathcal{L}_{\text{mixcon}}$ is applied to the 2nd fully-connected layer.

We train the network for 20 epochs with cross-entropy loss and SGD optimizer with 0.1 learning rate. We apply noise to the labels by randomly flipping $5\%$ of labels to increase training difficulty.

**Testing setup.** We compare the results under the following settings:

- Vanilla: training using only $\mathcal{L}_{\text{class}}$.
- MixCon: training with MixCon loss with parameters $(\lambda, \beta)$[4].

---

[4] $\lambda$ is the coefficient of penalty and $\beta$ is balancing term for data separability.

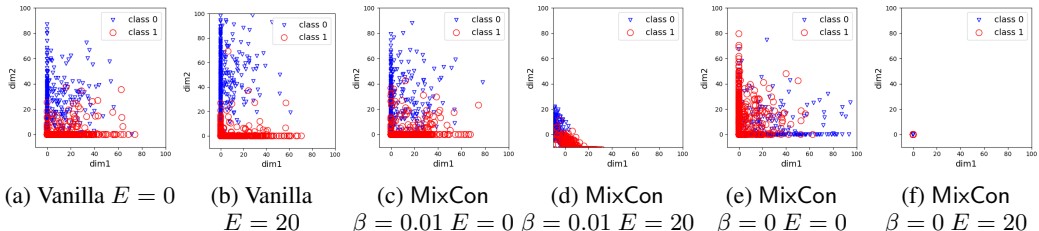

(a) Vanilla $E = 0$    (b) Vanilla $E = 20$    (c) MixCon $\beta = 0.01$ $E = 0$    (d) MixCon $\beta = 0.01$ $E = 20$    (e) MixCon $\beta = 0$ $E = 0$    (f) MixCon $\beta = 0$ $E = 20$

Figure 2: Data hidden representation $h(x) \in \mathbb{R}^2$ from the 2nd fully-connected layer of synthetic data at different epoch ($E$). Two settings of MixCon are given default $\lambda = 0.1$ but have different $\beta$. Compare to Vanilla, MixCon squeezes data representations to a smaller space over training. When $\beta = 0$, MixCon map all data to $h(x) = (0, 0)$, which is not learnable.

| | Vanilla | MixCon $\beta = 0.01$ | | | MixCon $\beta = 0$ | | |
|---|---|---|---|---|---|---|---|
| | | default | deeper net | wider net | default | deeper net | wider net |
| Train Accuracy (%) | 91.5 | 88.9 | 89.5 | 91.5 | 50.0 | 50.0 | 50.0 |
| Test Accuracy (%) | 91.5 | 88.0 | 88.5 | 90.5 | 50.0 | 50.0 | 50.0 |

Table 1: Data utility (accuracy). Vanilla is equivalent to ($\lambda = 0$, $\beta = 0$). Two MixCon "default" settings both use $\lambda = 0.1$ but vary in $\beta = 0.01$ and $\beta = 0$. "Deeper"/"Wider" indicate increasing the depth / width of layers in the network on server side $g(x)$.

We perform model inversion using Eq. (3) without any regularization term $\mathcal{R}(x)$ and $\mathcal{L}$ is the $\ell_1$-loss function. Detailed optimization process is listed in Appendix C.1.

**Results.** To answer Q1 that how $\beta$ in Eq.(1) affect the smallest data pairwise distance, we visualize the change of data representations at initial and ending epochs in Figure 2. First, in vanilla training (Figure 2 a-b), data are dispersively distributed and enlarge their distance after training. The obvious difference for MixCon training (Figure 2 c-f) is that data representations become more and more gathering through training. Second, we direct the data utility results of Vanilla and two "default" MixCon settings – ($\lambda = 0.1, \beta = 0.01$) and ($\lambda = 0.1, \beta = 0$) to Table 1. When $\beta = 0$, MixCon achieves chance accuracy only as it encodes all the $h(x)$ to hidden space (0,0) (Figure 2 f). While having $\beta > 0$ balancing the separability, MixCon achieves similar accuracy as Vanilla.

Based on Theorem 3.2, we further present two strategies to ensure reasonable accuracy while comprise of reducing data separability by increasing the depth or the width of the layers $g(z)$, the network after the layer that is applied $\mathcal{L}_{\mathrm{mixcon}}$. In practice, we add two more fully-connected layers with 100 neurons after the 3nd layer for "deeper" $g(x)$, and change the number of neurons on the 3nd layer to 2048 for "wider" $g(x)$. We show the utility results in Table 1. Using deeper or wider $g(z)$, MixCon ($\lambda = 0.1, \beta = 0.01$) improves accuracy. Whereas MixCon ($\lambda = 0.1, \beta = 0$) fails, because zero data separability is not learnable no matter how $g(z)$ changes. This gives conformable answer that $\beta$ is an important factor to guarantee neural network to be trainable.

To answer Q2, we evaluate the quality of data recovery using the inversion model. We use both square error (SE) and cosine similarity (CS) of $x$ and $x^*$ to evaluate the data recovery accuracy. We show the quantitative inversion results in Table 2 with the mean and worst case values. Higher SE or lower CS indicates a worse inversion. Apparently, data representation from MixCon trained network is more difficult to recover compared to Vanilla strategy.

| | Vanilla | MixCon (0.1, 0.01) | MixCon (0.1, 0) |
|---|---|---|---|
| SE | 1.92 (0.31) | 2.08 (0.37) | 2.35 (0.44) |
| CS | 0.169 (0.960) | 0.118 (0.939) | 0.161 (0.921) |

Table 2: Inversion results on synthetic dataset reported in mean(worst) format for the 200 testing samples. Higher MSE or lower MCS indicates a worse inversion. ($\lambda, \beta$) denoted in header.

### 4.3 EXPERIMENTS WITH BENCHMARK DATASETS

In this section, we would like to answer the following questions:

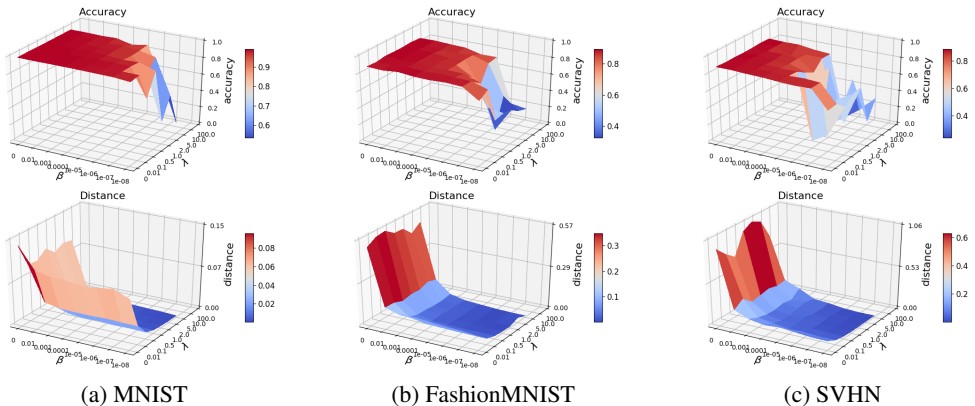

(a) MNIST      (b) FashionMNIST      (c) SVHN

Figure 3: Trade-off between data separability and data utility. We show testing accuracy and mean pairwise distance (data separability) on three datasets with different $\lambda$ and $\beta$. $\lambda$ and $\beta$ show complementary effort on adjusting data separability. A sweet-spot can be found at the $(\lambda, \beta)$ resulting in small data separability and high data utility.

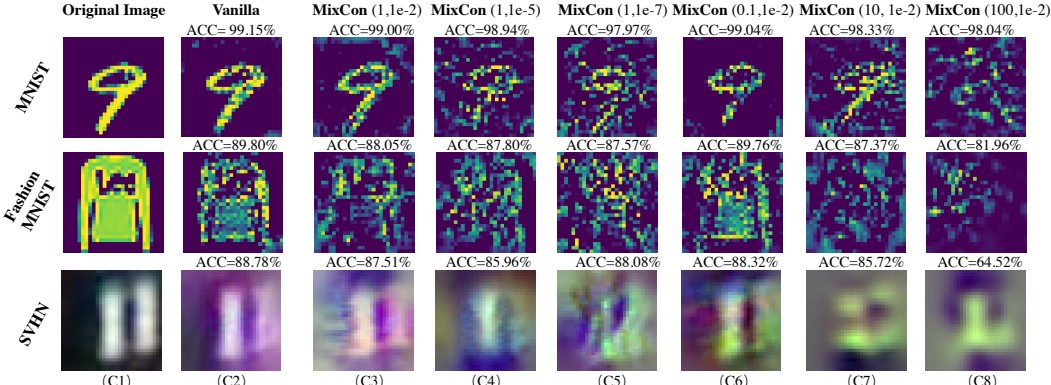

Figure 4: Qualitative evaluation for image inversion results. $(\lambda, \beta)$ settings of MixCon denoted on the header. The corresponding testing accuracy of each dataset is denoted on the top of each row. Compared to vanilla training, inversions from the MixCon model are less realistic and distinguishable from the original images without significant accuracy dropping.

Q3   How does MixCon loss affect data separability and accuracy on image datasets?

Q4   Are there parameters $(\lambda, \beta)$ in MixCon (Eq. (1)) to reach a "sweet-spot" for data utility and the quality of defending data recovery?

**Network, datasets and training setup.** The neural network architecture used in the experiments is LeNet5 LeCun (2015)[5]. $\mathcal{L}_{\mathrm{mixcon}}$ is applied to the outputs of the 2nd convolutional layer blocks of LeNet5. The experiments use three datasets: MNIST LeCun et al. (1998), Fashion-MNIST Xiao et al. (2017), and SVHN Netzer et al. (2011).

Neural network is optimized using cross-entropy loss and SGD optimizer with learning rate 0.01 for 20 epochs. We do not use any data augmentation or manual learning rate decay. MixCon loss is applied to the output of 2nd convolutional layer blocks in LeNet5. We use mini-batch training and each batch contains 40 data points from each class. [6] We train the model with different pairs of $(\lambda, \beta)$

---

[5]We change input channel to 3 for SVHN dataset.

[6]In the regular mini-batch training using data shuffling, when calculating the MixCon loss, we truncate the size of batch to $p|\mathcal{C}|$ and each mini-batch contains an equal number of samples of each class, Here $p$ is the number of training points of the smallest class and $|\mathcal{C}|$ is the number of classes.

| | MNIST | | FashionMNIST | | SVHN | |
|---|---|---|---|---|---|---|
| | Vanilla | MixCon | Vanilla | MixCon | Vanilla | MixCon |
| | - | $(\lambda = 1.0, \beta = 10^{-4})$ | - | $(\lambda = 1.0, \beta = 10^{-4})$ | - | $(\lambda = 0.5, \beta = 10^{-4})$ |
| Acc (%) | 99.1 | 98.6 | 89.8 | 88.9 | 88.4 | 88.2 |
| SSIM | $0.64 \pm 0.11(0.83)$ | $0.14 \pm 0.11(0.48)$ | $0.43 \pm 0.17(0.78)$ | $0.17 \pm 0.09(0.52)$ | $0.76 \pm 0.19(0.92)$ | $0.61 \pm 0.15(0.84)$ |
| PSIM | $0.78 \pm 0.05(0.88)$ | $0.44 \pm 0.07(0.69)$ | $0.71 \pm 0.13(0.92)$ | $0.42 \pm 0.08(0.66)$ | $0.69 \pm 0.07(0.81)$ | $0.59 \pm 0.07(0.72)$ |

Table 3: Quantitative evaluations for image recovery results. For fair evaluation, we match the data utility (accuracy) for Vanilla and MixCon. Structural similarity index metric (SSIM) and perceptual similarity (PSIM) are measured on 100 testing samples. Those scores are presented in mean $\pm$ std and worst-case (in parentheses) format. Lower scores indicate harder to invert.

in Eq. (1) for the following testing. Specifically, we vary $\lambda$ from: $\{0.01, 0.1, 0.5, 1, 2, 5, 10, 100\}$ and $\beta$ from: $\{10^{-2}, 10^{-3}, 10^{-4}, 10^{-5}, 10^{-6}, 10^{-7}, 10^{-8}\}$.

**Testing setup.** We record the testing accuracy and pairwise distance of data representation under each pair of $(\lambda, \beta)$ for each dataset. Following a recent model inversion method He et al. (2019), we define $\mathcal{L}$ in Eq. (3) as $\ell_2$-loss function, $\mathcal{R}$ as the regularization term capturing the total variation of a 2D signal defined as $\mathcal{R}(a) = \sum_{i,j}((a_{i+1,j} - a_{i,j})^2 + (a_{i,j+1} - a_{i,j})^2)^{1/2}$. The inversion attack is applied to the output of 2nd convolutional layer blocks in LeNet5 and find the optimal of Eq. (3) though SGD optimizer. Detailed optimization process is listed in Appendix C.2.

We use metrics normalized structural similarity index metric (SSIM) Wang et al. (2004) and perceptual similarity (PSIM) Johnson et al. (2016) to measure the similarity between the recovered image and the original image. The concrete definitions of SSIM and PSIM are listed in Appendix C.3.

**Results** To answer Q3, we plot the complementary effects of $\lambda$ and $\beta$ in Figure 3. Note that $\beta$ bounds the minimal pairwise of data representations, and $\lambda$ indicate the penalty power on data separability given by MixCon. Namely, a larger $\lambda$ brings stronger penalty of MixCon, which enhances the regularization of data separability and results in lower accuracy. Meanwhile, with a small $\beta$, $\lambda$ is not necessary to be very large, as smaller $\beta$ leads to a smaller bound of data separability, thus resulting in lower accuracy. Hence, $\lambda$ and $\beta$ work together to adjust the separability of hidden data representations, which can affect on data utility.

To answer Q4, we evaluate the quality of inversion qualitatively and quantitatively through a model inversion attack defined in "Test setup" paragraph. Specifically, for each private input $x$, we execute the inversion attack on $h_{\text{mixcon}}(x)$ and $h_{\text{vanilla}}(x)$ of testing images. As it is qualitatively shown in Figure 4, first, the recovered images using model inversion from MixCon training (such as given $(\lambda, \beta) \in \{(1, 1 \times 10^{-7}), (10, 1 \times 10^{-2}), (100, 1 \times 10^{-2})\}$) are visually different from the original inputs, while the recovered images from Vanilla training still look similar to the originals. Second, with the same $\lambda$ (Figure 4 column c3-c5), the smaller the $\beta$ it is, the less similar of the recovered images to original images. Last, with the same $\beta$ (Figure 4 column c3 and c6-c8), the larger the $\lambda$ it is, the less similar of the recovered images to original images.

Further, we quantitatively measure the inversion performance by reporting the averaged similarity between 100 pairs of recovered images by the inversion model and their original samples. We select $(\lambda, \beta)$ to match the accuracy [7] of MixCon with Vanilla training (see Accuracy in Table 3), and investigate if MixCon makes the inversion attack harder. The inverted results (see SSIM and PSIM in Table 3) are reported in the format of mean $\pm$ std and the worst case (the best-recovered data) similarity in parentheses for each metric. Both qualitative and quantitative results agree with our hypothesis that 1) adding $\mathcal{L}_{\text{mixcon}}$ in network training can reduce the mean pairwise distance (separability) of data hidden representations; and 2) smaller separability make it more difficult to invert original inputs. By sweet spot, we can define as the set of $(\beta, \lambda)$ that suffers with negligible accuracy loss (say within 1%) and the model inversion becomes significantly harder w.r.t computational complexity or breaks the attack (less similarity to the original input data per se). Thus by visiting through possible $(\lambda, \beta)$, we are able to find a spot, where data utility is reasonable but harder for data recovery, such as $(\lambda = 100, \beta = 1e - 2)$ for MNIST (Figure 4). Thus our proposed method is helpful if the user is willing to give up on some accuracy in the hope of getting a more robust model.

---

[7] Accuracy reduction is within a small tolerance, i.e., 1%.

## 5 DISCUSSION AND CONCLUSION

In this paper, we have proposed and studied the trade-off between data utility and data recovery from the angle of the separability of hidden data representations in deep neural network. We propose using MixCon, a consistency loss term, as an effective way to adjust the data separability. Our proposal is inspired by theoretical data separability results and a new exponential lower bound on approximately solving the network inversion problem, based on the Exponential Time Hypothesis (ETH).

We conduct two sets of experiments, using synthetic and benchmark datasets, to show the effect of adjusting data separability on accuracy and data recovery. Our theoretical insights help explain our key experimental findings: MixCon can effectively adjust the separability of hidden data representations, and one can find "sweet-spot" parameters for MixCon to make it difficult to recover data while maintaining data utility. Our experiments are limited to small benchmark datasets in the domain of image classifications. It will be helpful to conduct experiments using large datasets in multiple domains to further the study of the potential of adjusting data separability of data representations to trade-off between data utility and data recovery.

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

**Roadmap of Appendix** The Appendix is organized as follows. We discuss related work in Section A. We provide theoretical analysis in Section B. The details of data recovery experiment are in Section C and additional experiment details are in Section D.

# A    RELATED WORK

## A.1    HARDNESS AND NEURAL NETWORKS

When there are no further assumptions, neural networks have been shown hard in several different perspectives. Blum & Rivest (1992) first proved that learning the neural network is NP-complete. Different variant hardness results have been developed over past decades Klivans & Sherstov (2009); Daniely (2016); Daniely & Shalev-Shwartz (2016); Goel et al. (2017); Livni et al. (2014); Weng et al. (2018); Manurangsi & Reichman (2018); Lei et al. (2019); Daniely & Vardi (2020); Huang et al. (2020b;a). The work of Lei et al. (2019) is most relevant to us. They consider the neural network inversion problem in generative models and prove the exact inversion problem is NP-complete.

## A.2    DATA SEPARABILITY AND NEURAL NETWORK TRAINING

One popular distributional assumption, in theory, is to assume the input data points to be the Gaussian distributions Zhong et al. (2017b); Li & Yuan (2017); Zhong et al. (2017a); Ge et al. (2018); Bakshi et al. (2019); Chen et al. (2020) to show the convergence of training deep neural networks. Later, convergence analysis using weaker assumptions are proposed, i.e., input data points are separable Li & Liang (2018). Following Li & Liang (2018); Allen-Zhu et al. (2019b;c;a); Zhang et al. (2020a), data separability plays a crucial role in deep learning theory, especially in showing the convergence result of over-parameterized neural network training. Denote $\delta$ is the minimum gap between all pairs data points. Data separability theory says as long as the width ($m$) of neural network is at least polynomial factor of all the parameters ($m \geq \mathrm{poly}(n, d, 1/\delta)$), i.e., $n$ is the number of data points, $d$ is the dimension of data, and $delta$ is data separability. Another line of work Du et al. (2019); Arora et al. (2019a;b); Song & Yang (2019); Brand et al. (2020); Lee et al. (2020) builds on neural tangent kernel Jacot et al. (2018). It requires the minimum eigenvalue ($\lambda$) of neural tangent kernel is lower bounded. Recent work Oymak & Soltanolkotabi (2020) finds the connection between data-separabiblity $\delta$ and minimum eigenvalue $\lambda$, i.e. $\delta \geq \lambda/n^2$.

## A.3    DISTRIBUTED DEEP LEARNING SYSTEM

Collaboration between the edge device and cloud server achieves higher inference speed and lowers power consumption than running the task solely on the local or remote platform. Typically there are two collaborative modes. The first is collaborative training, for which training task is distributed to multiple participants Konečnỳ et al. (2015); Vanhaesebrouck et al. (2016); Kairouz et al. (2019). The other model is collaborative inference. In such a distributed system setting, the neural network can be divided into two parts. The first few layers of the network are stored in the local edge device, while the rest are offloaded to a remote cloud server. Given an input, the edge device calculates the output of the first few layers and sends it to the cloud. Then cloud perform the rest of computation and sends the final results to each edge device Eshratifar et al. (2019); Hauswald et al. (2014); Kang et al. (2017); Teerapittayanon et al. (2017). In our work, we focus on tackling data recovery problem under collaborative inference mode.

## A.4    MODEL INVERSION ATTACK AND DEFENSE

The neural network inversion problem has been extensively investigated in recent years Fredrikson et al. (2015); He et al. (2019); Lei et al. (2019); Zhang et al. (2020b). As used in this paper, the general approach is to cast the network inversion as an optimization problem and uses a problem specified objective. In particular, Fredrikson et al. (2015) proposes to use confidence in prediction as to the optimized objective. He et al. (2019) uses a regularized maximum likelihood estimation. Recent work Zhang et al. (2020b) also proposes to use GAN to do the model inversion.

There are very few studies about defenses against model inversion attack. Existing data privacy protection mechanisms mainly rely on noise injection Fredrikson et al. (2015); Dwork (2008); Abadi

et al. (2016) or Homomorphic Encryption Nandakumar et al. (2019). While being able to mitigate attacks, existing methods significantly hinder model performance. Recently MID Wang et al. (2020) was proposed to limit the information about the model input contained in the prediction, thereby limiting the ability of an adversary to infer data information from the model prediction. Yang et al. (2020) proposed to add a purification block following by prediction output, so that the confidence score vectors predicted by the target classifier are less sensitivity of the prediction to the change of input data. However, the above two methods target the logit output layer (i.e., performing argmax). They either require auxiliary information (i.e., knowing attack model) or modifying network structure (i.e., building variational autoencoder structure for mutual information calculation). In contrast, our proposed method MixConcan easily and efficiently serve as a plug-in loss to the middle layers of arbitrarily network structures to defend inversion attack during inference.

# B   HARDNESS OF NEURAL NETWORK INVERSION

## B.1   PRELIMINARIES

We first provide the definitions for 3SAT, ETH, MAX3SAT, MAXE3SAT and then state some fundamental results related to those definitions. For more details, we refer the reader to the textbook Arora & Barak (2009).

**Definition B.1** (3SAT problem). *Given $n$ variables and $m$ clauses in a conjunctive normal form* CNF *formula with the size of each clause at most* 3, *the goal is to decide whether there exists an assignment to the $n$ Boolean variables to make the* CNF *formula be satisfied.*

**Hypothesis B.2** (Exponential Time Hypothesis (ETH) Impagliazzo et al. (1998)). *There is a $\delta > 0$ such that the* 3SAT *problem defined in Definition B.1 cannot be solved in $O(2^{\delta n})$ time.*

ETH is a stronger notion than NP$\neq$ P, and is well acceptable the computational complexity community. Over the few years, there has been work proving hardness result under ETH for theoretical computer science problems Chalermsook et al. (2017); Manurangsi (2017); Chitnis et al. (2018); Bhattacharyya et al. (2018); Dinur & Manurangsi (2018); KCS & Manurangsi (2018) and machine learning problems, e.g. matrix factorizations Arora et al. (2012); Razenshteyn et al. (2016); Song et al. (2017); Ban et al. (2019), tensor decomposition Song et al. (2019). There are also variations of ETH, e.g. Gap-ETH Dinur (2016; 2017); Manurangsi & Raghavendra (2017) and random-ETH Feige (2002); Razenshteyn et al. (2016), which are also believable in the computational complexity community.

**Definition B.3** (MAX3SAT). *Given $n$ variables and $m$ clauses, a conjunctive normal form* CNF *formula with the size of each clause at most* 3, *the goal is to find an assignment that satisfies the largest number of clauses.*

We use MAXE3SAT to denote the version of MAX3SAT where each clause contains exactly 3 literals.

**Theorem B.4** (Håstad (2001)). *For every $\delta > 0$, it is* NP-*hard to distinguish a satisfiable instance of* MAXE3SAT *from an instance where at most a $7/8 + \delta$ fraction of the clauses can be simultaneously satisfied.*

**Theorem B.5** (Håstad (2001); Moshkovitz & Raz (2010)). *Assume* ETH *holds. For every $\delta > 0$, there is no $2^{o(n^{1-o(1)})}$ time algorithm to distinguish a satisfiable instance of* MAXE3SAT *from an instance where at most a fraction $7/8 + \delta$ of the clauses can be simultaneously satisfied.*

We use MAXE3SAT(B) to denote the restricted special case of MAX3SAT where every variable occurs in at most $B$ clauses. Håstad Håstad (2000) proved that the problem is approximable to within a factor $7/8 + 1/(64B)$ in polynomial time, and that it is hard to approximate within a factor $7/8 + 1/(\log B)^{\Omega(1)}$. In 2001, Trevisan improved the hardness result,

**Theorem B.6** (Trevisan (2001)). *Unless* RP=NP, *there is no polynomial time $(7/8 + 5/\sqrt{B})$-approximate algorithm for* MAXE3SAT(B).

**Theorem B.7** (Håstad (2001); Trevisan (2001); Moshkovitz & Raz (2010)). *Unless* ETH *fails, there is no $2^{o(n^{1-o(1)})}$ time $(7/8 + 5/\sqrt{B})$-approximate algorithm for* MAXE3SAT(B).

### B.2 OUR RESULTS

We provide a hardness of approximation result for the neural network inversion problem. In particular, we prove unless RP=NP, there is no polynomial time that can approximately recover the input of a two-layer neural network with ReLU activation function[8]. Formally, consider the inversion problem

$$h(x) = z, \quad x \in [-1, 1]^d, \tag{4}$$

where $z \in \mathbb{R}^{m_2}$ is the hidden layer representation, $h$ is a two neural network with ReLU gates, specified as

$$h(x) = W_2\sigma(W_1 x + b), \quad W_2 \in \mathbb{R}^{m_2 \times m_1}, W_1 \in \mathbb{R}^{m_1 \times d}, b \in \mathbb{R}^{m_1}$$

We want to recover the input data $x \in [-1, 1]^d$ given hidden layer representation $z$ and all parameters of the neural network (i.e., $W^{(1)}, W^{(2)}, b$). It is known the decision version of neural network inversion problem is NP-hard Lei et al. (2019). It is an open question whether approximation version is also hard. We show a stronger result which is, it is hard to give to constant approximation factor. Two notions of approximation could be consider here, one we called *solution approximation*

**Definition B.8** (Solution approximation)**.** *Given a neural network $h$ and hidden layer representation $z$, we say $x' \in [-1, 1]^d$ is an $\epsilon$ approximation solution for Eq. (4), if there exists $x \in [-1, 1] \in \mathbb{R}^d$, such that*

$$\|x - x'\|_2 \leq \epsilon\sqrt{d} \text{ and } h(x) = z.$$

Roughly speaking, solution approximation means we recovery an approximate solution. The $\sqrt{d}$ factor in the above definition is a normalization factor and it is not essential.

One can also consider a weaker notion, which we called *function value approximation*

**Definition B.9** (Function value approximation)**.** *Given a neural network $h$ and hidden layer representation $z$, we say $x' \in [-1, 1]^d$ is $\epsilon$-approximate of value to Eq. (4), if*

$$\|h(x') - y\|_2 \leq \epsilon\sqrt{m_2}.$$

Again, the $\sqrt{m_2}$ factor is only for normalization. Suppose the neural network is $G$-Lipschitz continuous for constant $G$ (which is the case in our proof), then an $\epsilon$-approximate solution implies $G\epsilon$-approximation of value. For the purpose of this paper, we focus on the second notion (i.e., function value approximation). Given our neural network is (constant)-Lipschitz continuous, this immediately implies hardness result for the first one.

Our theorem is formally stated below. In the proof, we reduce from MAX3SAT($B$) and utilize Theorem B.6

**Theorem B.10.** *There exists a constant $B > 1$, unless RP = NP, it is hard to $\frac{1}{60B}$-approximate Eq. (4) . Furthermore, the neural network is $O(B)$-Lipschitz continuous, and therefore, it is hard to find an $\Omega(1/B^2)$ approximate solution to the neural network.*

Using the above theorem, we can see that by taking a suitable constant $B > 1$, the neural network inversion problem is hard to approximate within some constant factor under both definitions. In particular, we conclude

**Theorem B.11** (Formal statement of Theorem 3.3)**.** *Assume $NP \neq RP$, there exists a constant $\epsilon > 0$, such that there is no polynomial time algorithm that is able to give an $\epsilon$-approximation to neural network inversion problem.*

*Proof of Theorem B.10.* Given an 3SAT instance $\phi$ with $n$ variables and $m$ clause, where each variable appears in at most $B$ clauses, we construct a two layer neural network $h_\phi$ and output representation $z$ satisfy the following:

- Completeness. If $\phi$ is satisfiable, then there exists $x \in [0, 1]^d$ such that $h_\phi(x) = z$.

---

[8]We remark there is a polynomial time algorithm for one layer ReLU neural network recovery

- Soundness. For any $x$ such that $\|h_\phi(x) - z\|_2 \le \frac{1}{60B}\sqrt{m_2}$, we can recover an assignment to $\phi$ that satisfies at least $\left(\frac{7}{8} + \frac{5}{\sqrt{B}}\right) m$ clauses

- Lipschitz continuous. The neural network is $O(B)$-Lipschitz.

We set $d = n$, $m_1 = m + 200B^2 n$ and $m_2 = m + 100B^2 n$. For any $j \in [m]$, we use $\phi_j$ to denote the $j$-th clause and use $h_{1,j}(x)$ to denote the output of the $j$-th neuron in the first layer, i.e., $h_{1,j}(x) = \sigma(W_j^{(1)} x + b_i)$, where $W_j^{(1)}$ is the $j$-th row of $W^{(1)}$. For any $i \in [n]$, we use $X_i$ to denote the $i$-th variable.

Intuitively, we use the input vector $x \in [-1, 1]^n$ to denote the variable, and the first $m$ neurons in the first layer to denote the $m$ clauses. By taking

$$W_{j,i}^{(1)} = \begin{cases} 1, & X_i \in \phi_j; \\ -1, & \overline{X}_i \in \phi_j; \\ 0, & \text{otherwise.} \end{cases} \quad \text{and} \quad b_j = -2$$

for any $i \in [n], j \in [m]$, and viewing $x_i = 1$ as $X_i$ to be false and $x_i = -1$ as $X_i$ to be true. One can verify that $h_{1,j}(x) = 0$ if the clause is satisfied, and $h_{1,j}(x) = 1$ if the clause is unsatisfied. We simply copy the value in the second layer $h_j(x) = h_{1,j}(x)$ for $j \in [m]$.

For other neurons, intuitively, we make $100B^2$ copies for each $|x_i|$ $(i \in n)$ in the output layer. This can be achieved by taking

$$h_{m+(i-1)\cdot 100B^2 + k}(x) = h_{m+(i-1)\cdot 100B^2 + k}(x) + h_{1,m+100B^2 n+(i-1)\cdot 100B^2 + k}(x)$$

and set

$$h_{1,m+(i-1)\cdot 100B^2 + k}(x) = \max\{x_i, 0\} \quad h_{1,m+100B^2 n+(i-1)\cdot 100B^2 + k}(x) = \max\{-x_i, 0\}$$

for any $i \in [n], k \in [100B^2]$. Finally, we set the target output as

$$z = (\underbrace{0, \cdots, 0}_{m}, \underbrace{1, \cdots, 1}_{100B^2 n})$$

We are left to prove the three claims we made about the neural network $h$ and the target output $z$. For the first claim, suppose $\phi$ is satisfiable and $X = (X_1, \cdots, X_n)$ is the assignment. Then as argued before, we can simply take $x_i = 1$ if $X_i$ is false and $x_i = -1$ is $X_i$ is true. One can check that $h(x) = z$.

For second claim, suppose we are given $x \in [-1, 1]^d$ such that

$$\|h(x) - z\|_2 \le \frac{1}{60B}\sqrt{m_2}$$

We start from the simple case when $x$ is binary, i.e., $x \in \{-1, 1\}^n$. Again, by taking $X_i$ to be true if $x_i = -1$ and $X_i$ to be false when $x_i = 0$. One can check that the number of unsatisfied clause is at most

$$\begin{aligned}
\|h(x) - z\|_2^2 &\le \frac{1}{3600B^2} m_2 \\
&= \frac{1}{3600B^2}(m + 100B^2 n) \\
&\le \frac{1}{12}m + \frac{1}{3600B^2}m \\
&\le \frac{1}{8}m - \frac{5}{\sqrt{B}}m
\end{aligned} \tag{5}$$

The third step follows from $n \le 3m$, and the last step follows from $B \ge 15000$.

Next, we move to the general case that $x \in [-1, 1]^d$. We would round $x_i$ to $-1$ or $+1$ based on the sign. Define $\overline{x} \in \{-1, 1\}^n$ as

$$\overline{x}_i = \arg \min_{t \in \{-1,1\}} |t - x_i|$$

We prove that $\overline{x}$ induces an assignment that satisfies $(\frac{7}{8} + \frac{5}{\sqrt{B}})m$ clauses. It suffices to prove

$$\|h(\overline{x}) - z\|_2^2 - \|h(x) - z\|_2^2 \le \frac{3}{100}m \tag{6}$$

since this implies the number of unsatisfied clause is bounded by

$$\|h(\overline{x}) - z\|_2^2 \le \|h(x) - z\|_2^2 + (\|h(\overline{x}) - z\|_2^2 - \|h(x) - z\|_2^2)$$
$$\le (\frac{1}{12}m + \frac{1}{36B^2}m) + \frac{3}{100}m$$
$$\le \frac{1}{8}m - \frac{1}{5\sqrt{B}}m,$$

where the second step follow from Eq. (5)(6), and the last step follows from $B \ge 10^7$.

We define $\Delta_i := |\overline{x}_i - x_i| = 1 - |x_i| \in [0, 1]$ and $T := m + 128B^2n$. Then we have

$$\|h(\overline{x}) - z\|_2^2 - \|h(\overline{x}) - z\|_2^2 = \sum_{j=1}^{T}(h_j(\overline{x}) - z_j) - (h_j(x) - z_j)^2$$
$$= \sum_{j=1}^{m}(h_j(\overline{x}) - z_j)^2 - (h_j(x) - z_j)^2$$
$$+ \sum_{j=m+1}^{T}(h_j(\overline{x}) - z_j)^2 - (h_j(x) - z_j)^2$$
$$= \sum_{j=1}^{m} h_j(\overline{x})^2 - h_j(x)^2 - 100B^2 \sum_{i=1}^{n} \Delta_i^2$$
$$\le 2\sum_{j=1}^{m} |h_{1,j}(\overline{x}) - h_{1,j}(x)| - 100B^2 \sum_{i=1}^{n} \Delta_i^2$$
$$\le 2\sum_{j=1}^{m} \sum_{i \in \phi_j} \Delta_i - 100B^2 \sum_{i=1}^{n} \Delta_i^2$$
$$\le 2B\sum_{i=1}^{n} \Delta_i - 100B^2 \sum_{i=1}^{n} \Delta_i^2$$
$$\le \frac{n}{100}$$
$$\le \frac{3m}{100}.$$

The third step follow from $z_j = 0$ for $j \in [m]$ and for $j \in \{m+1, \cdots, m + 100B^2n\}$, $z_j = 1$, $\|h_j(\overline{x}) - z_j\| = 0$ and $\|h_j(x) - z_j\|_2^2 = \Delta_i$ given $j \in [m + (i-1) \cdot 100B^2 + 1, i \cdot 100B^2]$. The fourth step follows from that $h_j(x) = h_{1,j}(x) \in [0, 1]$ for $j \in [m]$. The fifth step follows from the 1-Lipschitz continuity of the ReLU. The sixth step follows from each variable appears in at most $B$ clause. This concludes the second claim.

For the last claim, by the Lipschitz continuity of ReLU, we have for any $x_1, x_2$

$$h(x_1) - h(x_2) = W^{(2)}\sigma(W^{(1)}x_1 + b) - W^{(2)}\sigma(W^{(1)}x_2 + b)$$
$$\le \|W^{(2)}\| \cdot \|W^{(1)}\|\|x_1 - x_2\|_2$$

It is easy to see that

$$\|W^{(2)}\| \le 2$$

and

$$\|W^{(2)}\| \le \sqrt{200B^2 + 3B} \le \sqrt{203B^2} \le 15B,$$

where the second step follows from $B \geq 1$.

Thus concluding the proof.

$\square$

By assuming ETH and using Theorem B.7, we can conclude

**Corollary B.12** (Formal statement of Corollary 3.5). *Unless ETH fails, there exists a constant $\epsilon > 0$, such that there is no $2^{o(n^{1-o(1)})}$ time algorithm that is able to give an $\epsilon$-approximation to neural network inversion problem.*

The proof is similar to Theorem B.10, we omit it here.

## C    DETAILS OF DATA RECOVERY EXPERIMENTS

### C.1    INVERSION MODEL DETAILS FOR SYNTHETIC DATASET

In synthetic experiment, a malicious attacker recover original input data $x \in \mathbb{R}^d$ by solving the the following optimization:

$$x^* = \arg\min_{s \in \mathbb{R}^d} \|h(s) - z\|_1$$

To estimate the optimal, we run an SGD optimizer with a learning rate of 0.01 and decayed weight $10^{-4}$ for 500 iterations. We test data recovery results on all the 200 testing samples. Namely, we solve the above optimization problems 200 times. Each time for a testing data point.

### C.2    INVERSION MODEL DETAILS FOR BENCHMARK DATASET

In benchmark experiment, a malicious attacker recover original input data $x \in \mathbb{R}^d$ by solving the the following optimization:

$$x^* = \arg\min_{s \in \mathbb{R}^d} \|h(s) - z\|_2 + \zeta \sum_{i,j}((s_{i+1,j} - s_{i,j})^2 + (s_{i,j+1} - s_{i,j})^2)^{1/2},$$

where $i, \ j$ are the indexes of pixels in an image.

To estimate the optimal, we run an SGD optimizer with a learning rate of 10 and decayed weight $10^{-4}$ for 500 iterations. We used a grid searching on the space of $\zeta$. We find that the best data recovery comes from $\zeta = 0.01$ for SVHN dataset and $\zeta = 10^{-5}$ for MNIST and FashionMNIST by grid search.

### C.3    QUANTITATIVE METRICS FOR IMAGE SIMILARITY MEASUREMENT

We adopt the following two known metrics to measure the similarity between $x^*$ and $x$:

- Normalized structural similarity index metric (**SSIM**), a perception-based metric that considers the similarity between images in structural information, luminance and contrast. It is widely used in image and video compression research to quantify the difference between the original and compressed images. The detailed calculation can be found in Wang et al. (2004). We normalize SSIM to take value range $[0, 1]$ (original SSIM takes value range $[-1, 1]$).

- Perceptual similarity (**PSIM**). Perceptual loss Johnson et al. (2016) has been widely used for training image generation and style transferring models Johnson et al. (2016); Lucas et al. (2019); Wang et al. (2018). It emerges as a novel measurement for evaluating the discrepancy between high-level perceptual features that extracted by deep learning model of the reconstructed image and ground-truth image. We define PSIM as $1-$ perceptual loss.

# D ADDITIONAL EXPERIMENTAL RESULTS

## D.1 COMPARE PENALTY STRATEGIES

A natural approach arise to reduce data separability could be adding a penalty on the pair-wise distance for the data representations within a class. We name this approach as UniCon. Its loss function denoted as $\mathcal{L}_{\text{unicon}}$ can be written as:

$$\mathcal{L}_{\text{unicon}} = \frac{1}{C} \frac{1}{|\mathcal{C}_c| \cdot (|\mathcal{C}_c| - 1)} \sum_{c \in \mathcal{C}} \sum_{i \in \mathcal{C}_c} \sum_{j \in \mathcal{C}_c} \|h(x_i) - h(x_j)\|_2^2,$$

The final objective function $\mathcal{L} := \mathcal{L}_{\text{class}} + \lambda \cdot \mathcal{L}_{\text{unicon}}$. This approach is similar to contrastive learning Khosla et al. (2020). However, we observed that the approach is not as ideal as our proposed MixCon, in the sense of defending inversion attack. The intuition is that MixCon can induce confusing patterns to fool the neural network learning typical patterns from a class. Here we show the visualization for the three benchmark datasets in Figure 5. We select $\lambda = 1$ for MNIST and FashionMNIST and $\lambda = 0.5$ for SVHN in both UniCon and MixCon. Then we choose the $\beta = 1e - 4$ for MixCon to match the accuracy to Vanilla and UniCon. We use the same training and testing of MixCon for UniCon experiment. From the representative samples (while typical to the rest of the data samples), we observe worse data recovery quality of MixCon. Notably, the recovered results from UniCon keep the pattern of their class. While MixCon results in more blurred and indistinguishable patterns across classes. We compare the quantitative evaluation results between MixCon and UniCon in Table 4. [9] We use metric SSIM and PSIM to evaluate the similarity between the recovered image and the original image. Lower scores indicate worse data recovery results. The data recovery experiment is performed on 100 testing samples, and we report the mean ± std and worst case (the best-recovered data) results. Except for the PSIM scores evaluated on MNIST, we get conformable evidence showing MixCon training is apt to defend inversion.

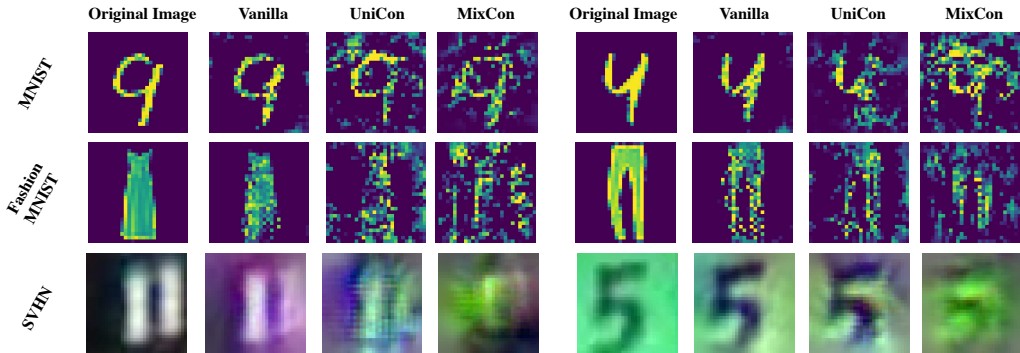

Figure 5: Qualitative evaluation for image inversion results.

| | MNIST | | FashionMNIST | | SVHN | |
|---|---|---|---|---|---|---|
| | UniCon | MixCon | UniCon | MixCon | UniCon | MixCon |
| | $\lambda = 1.0$ | $(\lambda = 1.0, \beta = 10^{-4})$ | $\lambda = 1.0$ | $(\lambda = 1.0, \beta = 10^{-4})$ | $\lambda = 0.5$ | $(\lambda = 0.5, \beta = 10^{-4})$ |
| Acc (%) | 99.2 | 98.6 | 89.6 | 88.9 | 88.3 | 88.2 |
| SSIM | $0.31 \pm 0.11(0.59)$ | $0.14 \pm 0.11(0.48)$ | $0.19 \pm 0.09(0.53)$ | $0.17 \pm 0.09(0.52)$ | $0.67 \pm 0.11(0.91)$ | $0.61 \pm 0.15(0.84)$ |
| PSIM | $0.41 \pm 0.07(0.60)$ | $0.44 \pm 0.07(0.69)$ | $0.45 \pm 0.07(0.64)$ | $0.42 \pm 0.08(0.66)$ | $0.62 \pm 0.05(0.75)$ | $0.59 \pm 0.07(0.72)$ |

Table 4: Quantitative evaluations for image recovery results. For fair evaluation, we match the data utility (accuracy) for Vanilla and MixCon. SSIM and PSIM are measured on 100 testing samples. Those scores are presented in mean ± std and worst-case (in parentheses) format. The smaller scores indicate harder data recovery.

---

[9]We have presented the comparison between MixCon and vanilla training in Table 3.

## D.2   Effects on the Selection of Middle Layers

The trade-off between data separability and data utility can be different for adding MixConon the different layers. In our benchmark experiment, we use a LeNet5 LeCun (2015) — a five-layer CNN. Thus there are four split methods, namely four intermediate outputs. In our collaborative inference setting [10], we visit the all four possible middle layers to apply MixCon loss. We plot the accuracy and data separability plots over the different combinations of $(\beta, \lambda)$, together with SSIM and PSIM scores (mean and the worst-case results), for each layer on our three benchmark datasets. The results are shown as Figure 6 to Figure 17. There is a clear trend that the shallower the $h(x)$ it is, the easier to recover original $x$ on average with respect to the mean SSIM and PSIM scores. The worst-case measurement may suffer from some outliers and imperfectness of the evaluation metrics. In most cases, distance and recovery similarity score for the first three layers shows a positive relationship, i.e. in Figure 6 - Figure 8. Usually, inversion from the deeper layers is not stable. Also, splitting a network at a deeper layer in the collaborative inference setting is not common or realistic because clients, such as edge-end devices, do not have powerful computational resources. Notably, the relationship between accuracy and similarity is highly non-linear. The sweet spot for a trade-off between accuracy and difficulty of recovery is in the space where the accuracy degradation curve is slow, while recovery similarity is low. Users can search the best parameters and "cut layer" to meet certain accuracy and data recovery defending requirements in practice.

---

[10]There is no necessity to add MixCon loss for the layers before "cut layer", because the attacker is not able to get access to the original data hidden representations form those layers

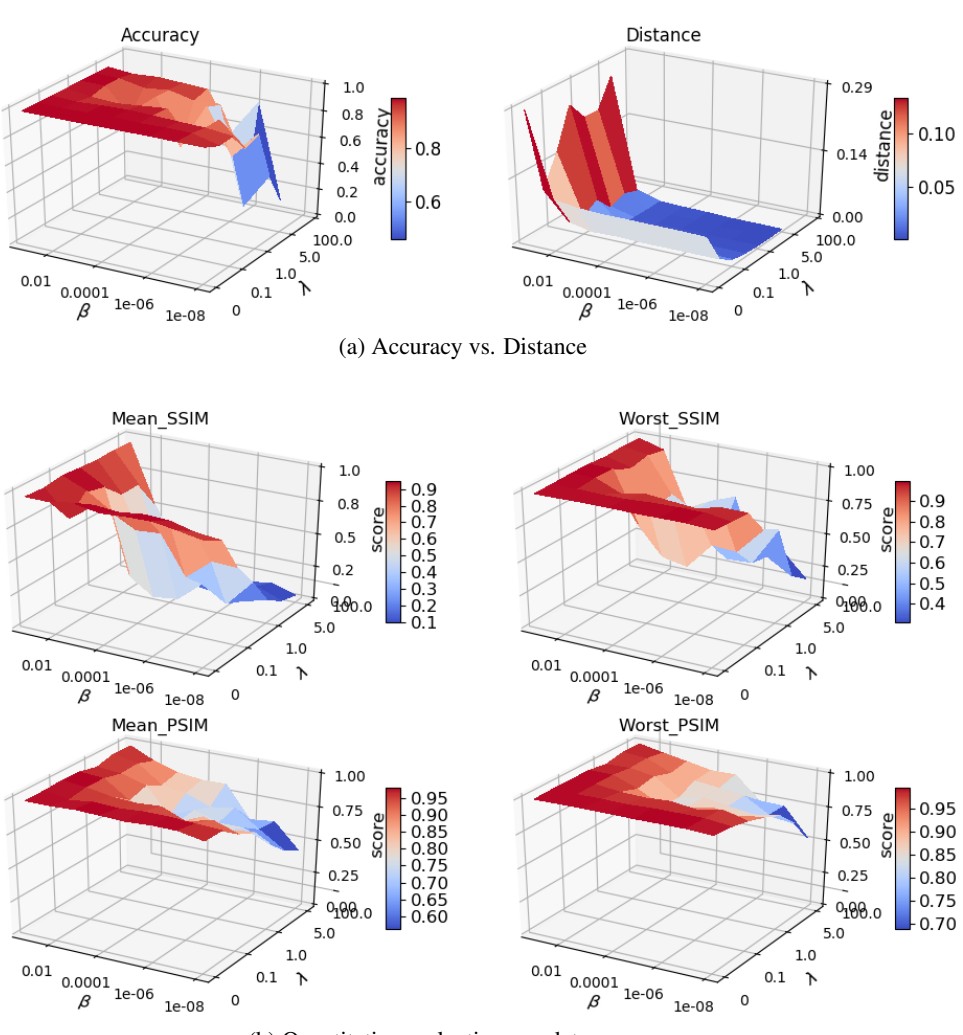

(a) Accuracy vs. Distance

(b) Quantitative evaluations on data recovery

Figure 6: Adding MixCon to the 1st layer of CNN on MNIST dataset. (a) The trade-off between data separability and data utility . We show testing accuracy and mean pairwise distance (data separability) with different $\lambda$ and $\beta$. $\lambda$ and $\beta$ show complementary effort on adjusting data separability. (b) Quantitative evaluation of data recovery results. We show SSIM and PSIM scores with different $\lambda$ and $\beta$.

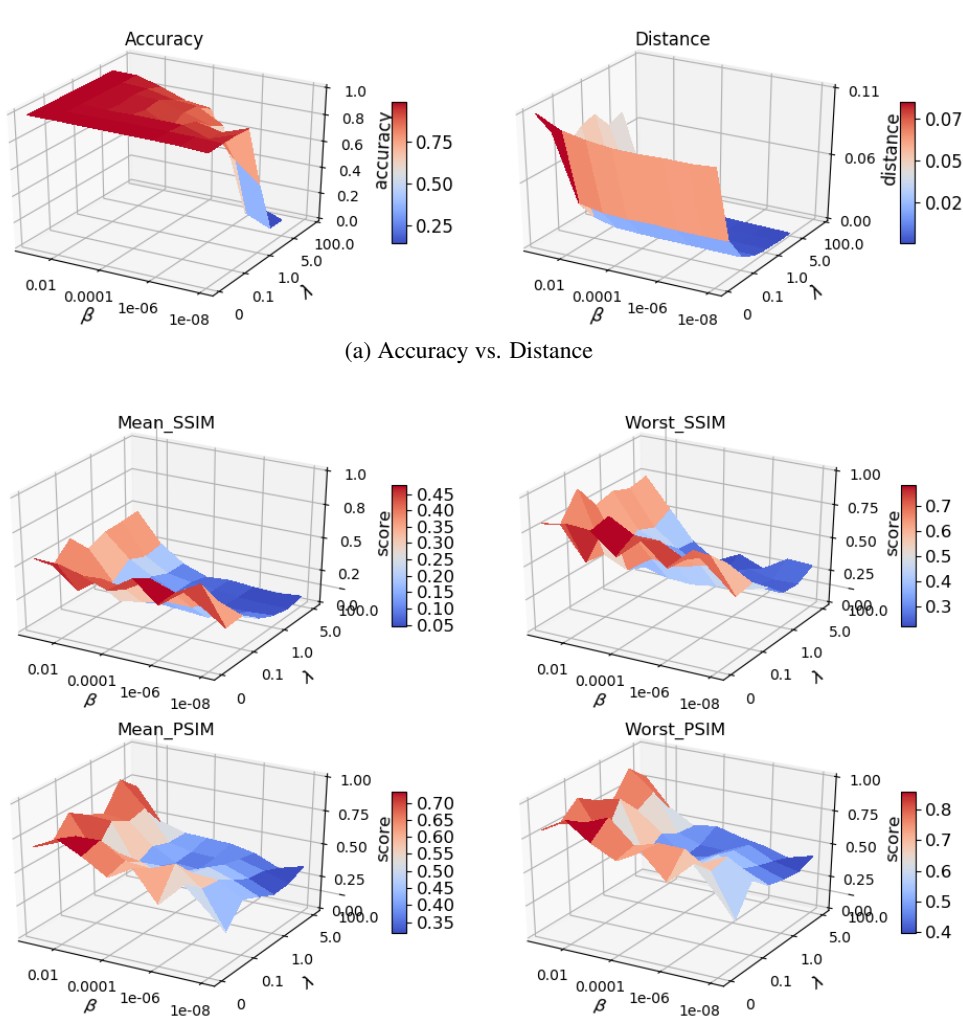

(a) Accuracy vs. Distance

(b) Quantitative evaluations on data recovery

Figure 7: Adding MixCon to the 2nd layer of CNN on MNIST dataset. (a) The trade-off between data separability and data utility . We show testing accuracy and mean pairwise distance (data separability) with different $\lambda$ and $\beta$. $\lambda$ and $\beta$ show complementary effort on adjusting data separability. (b) Quantitative evaluation of data recovery results. We show SSIM and PSIM scores with different $\lambda$ and $\beta$.

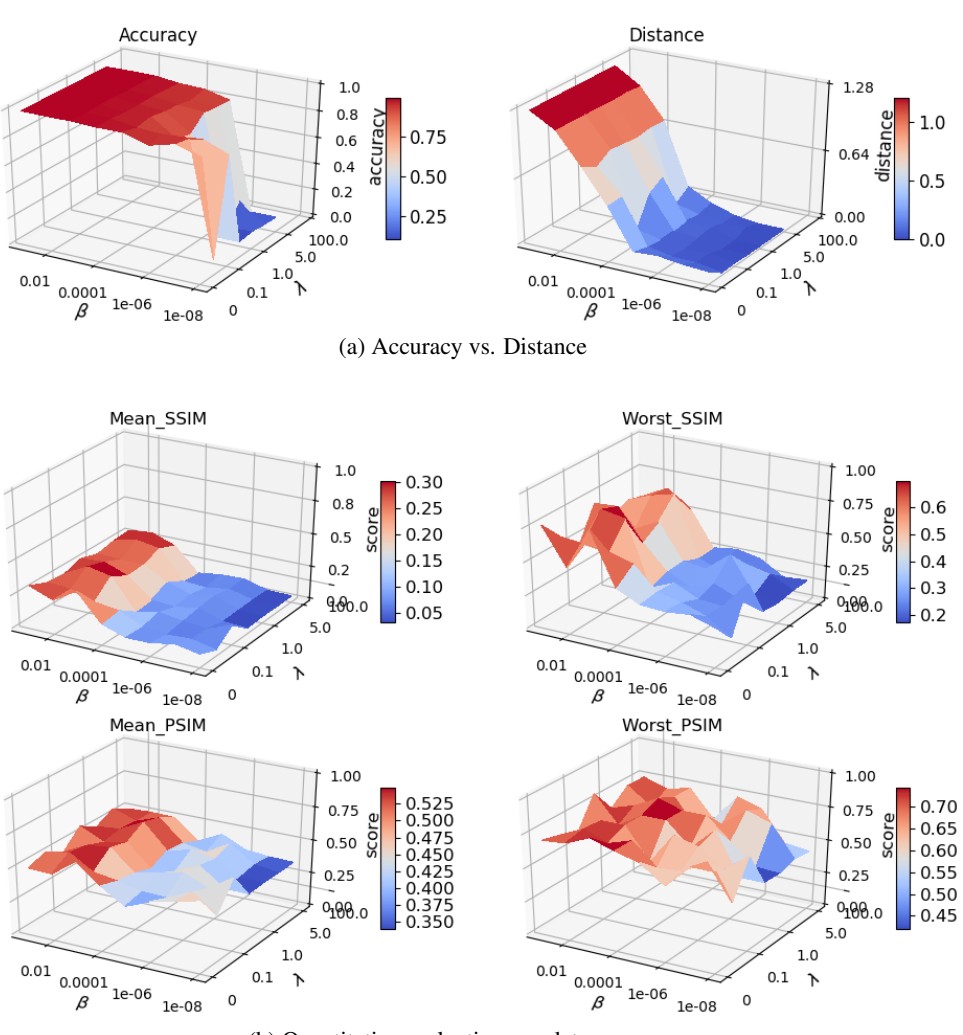

(a) Accuracy vs. Distance

(b) Quantitative evaluations on data recovery

Figure 8: Adding MixCon to the 3rd layer of CNN on MNIST dataset. (a) The trade-off between data separability and data utility . We show testing accuracy and mean pairwise distance (data separability) with different $\lambda$ and $\beta$. $\lambda$ and $\beta$ show complementary effort on adjusting data separability. (b) Quantitative evaluation of data recovery results. We show SSIM and PSIM scores with different $\lambda$ and $\beta$.

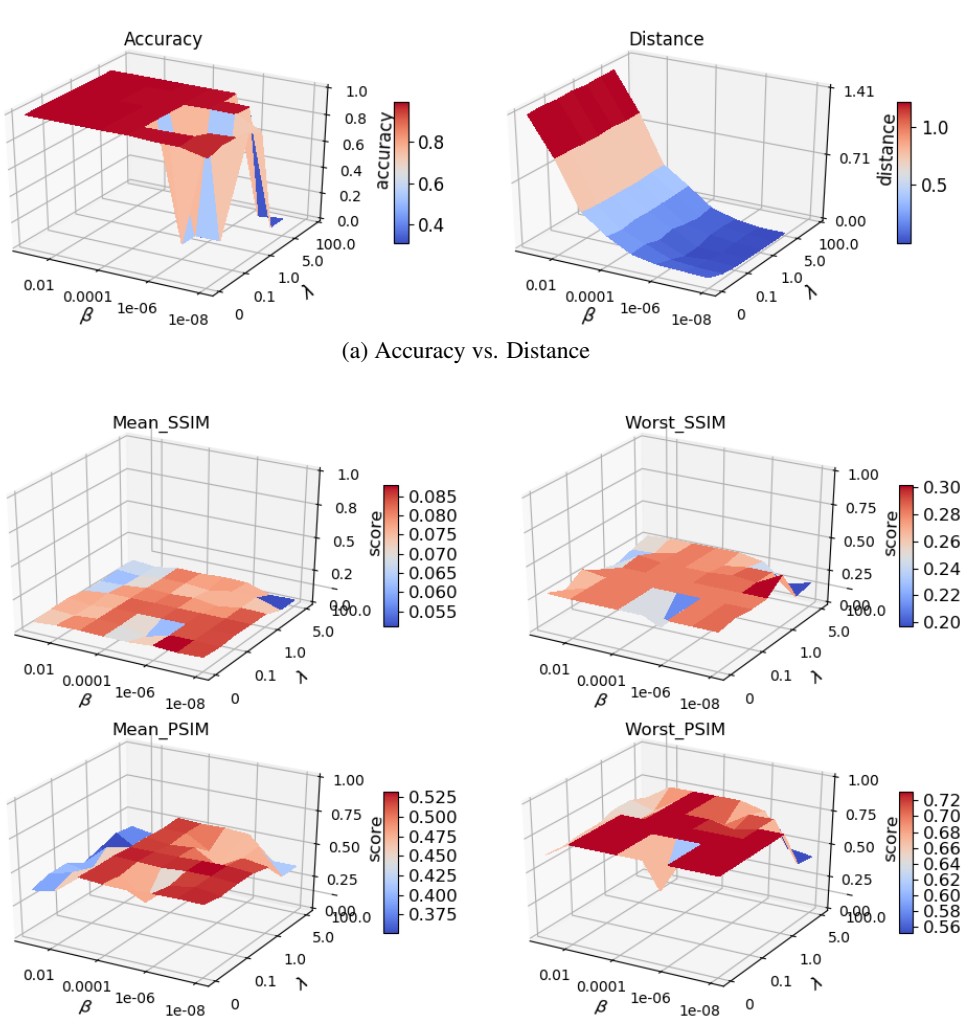

(a) Accuracy vs. Distance

(b) Quantitative evaluations on data recovery

Figure 9: Adding MixCon to the 4th layer of CNN on MNIST dataset. (a) The trade-off between data separability and data utility . We show testing accuracy and mean pairwise distance (data separability) with different $\lambda$ and $\beta$. $\lambda$ and $\beta$ show complementary effort on adjusting data separability. (b) Quantitative evaluation of data recovery results. We show SSIM and PSIM scores with different $\lambda$ and $\beta$.

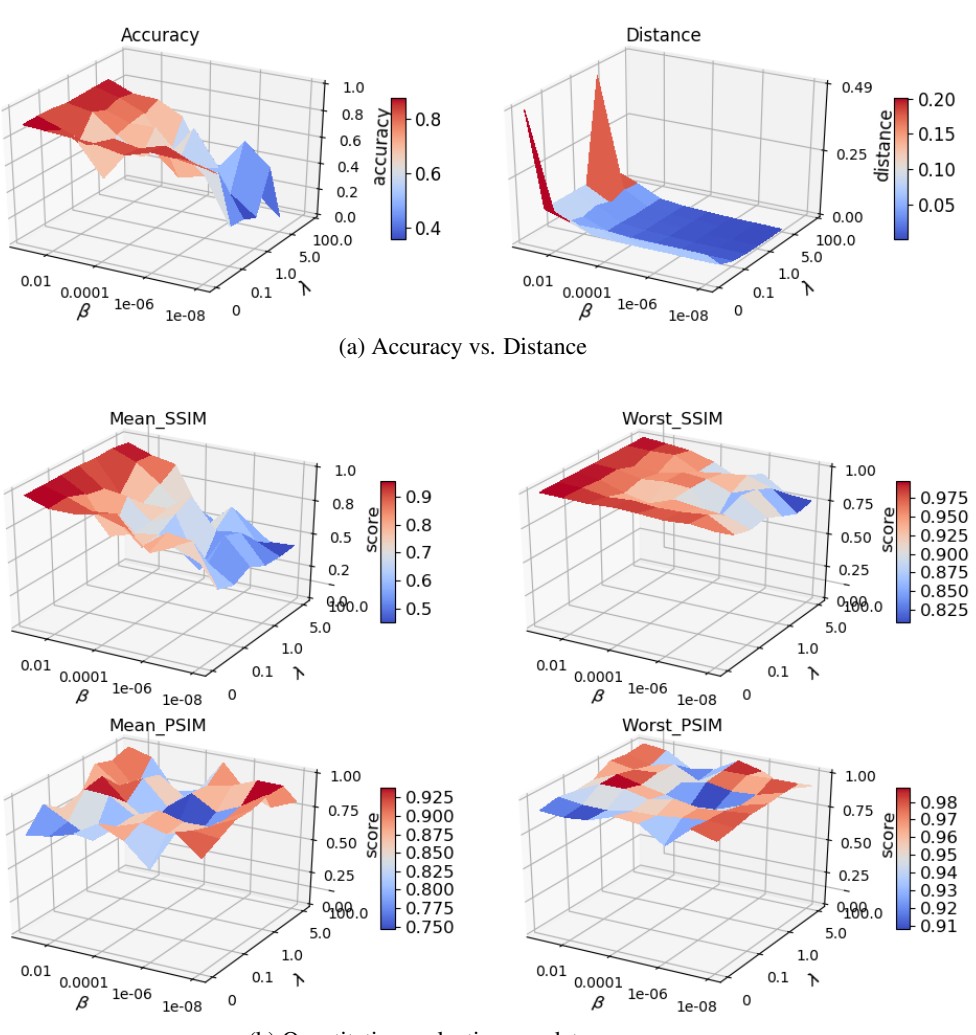

(a) Accuracy vs. Distance

(b) Quantitative evaluations on data recovery

Figure 10: Adding MixCon to the 1st layer of CNN on FashionMNIST dataset. (a) The trade-off between data separability and data utility . We show testing accuracy and mean pairwise distance (data separability) with different $\lambda$ and $\beta$. $\lambda$ and $\beta$ show complementary effort on adjusting data separability. (b) Quantitative evaluation of data recovery results. We show SSIM and PSIM scores with different $\lambda$ and $\beta$.

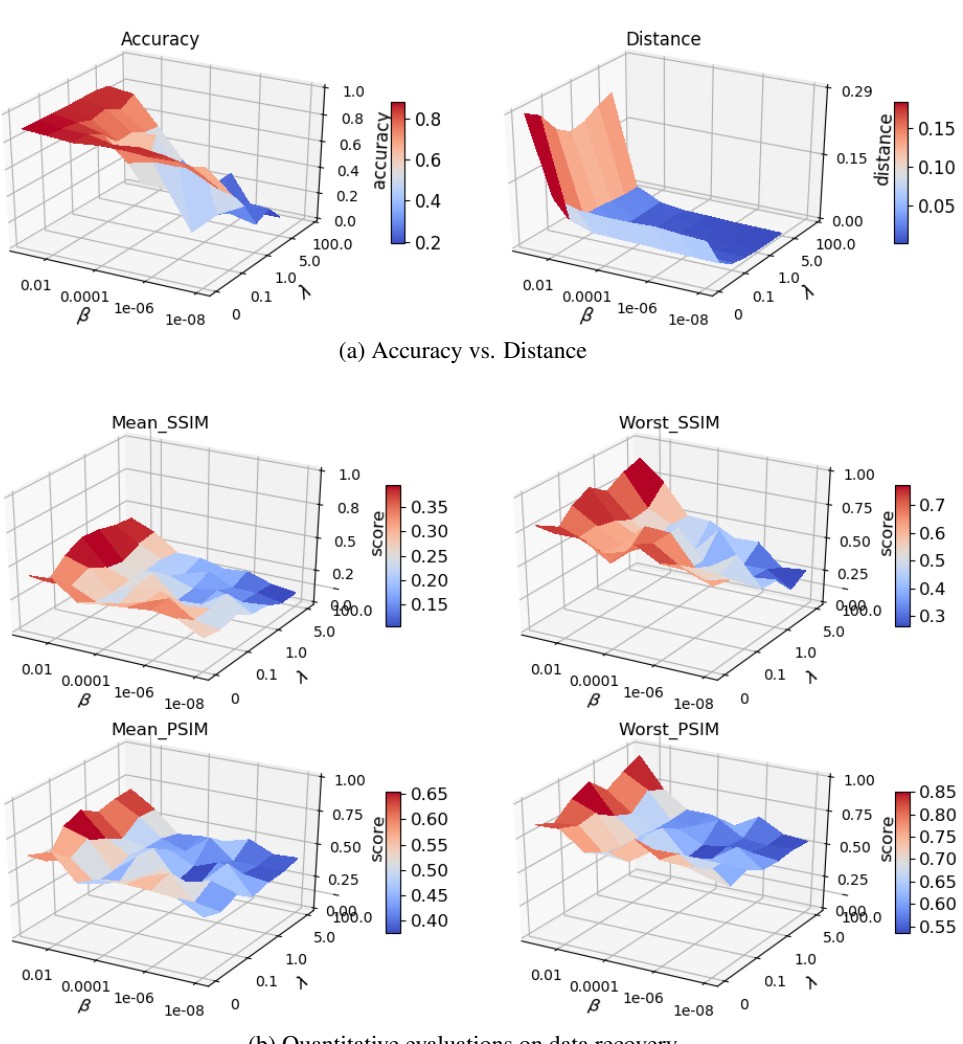

(a) Accuracy vs. Distance

(b) Quantitative evaluations on data recovery

Figure 11: Adding MixCon to the 2nd layer of CNN on FashionMNIST dataset. (a) The trade-off between data separability and data utility . We show testing accuracy and mean pairwise distance (data separability) with different $\lambda$ and $\beta$. $\lambda$ and $\beta$ show complementary effort on adjusting data separability. (b) Quantitative evaluation of data recovery results. We show SSIM and PSIM scores with different $\lambda$ and $\beta$.

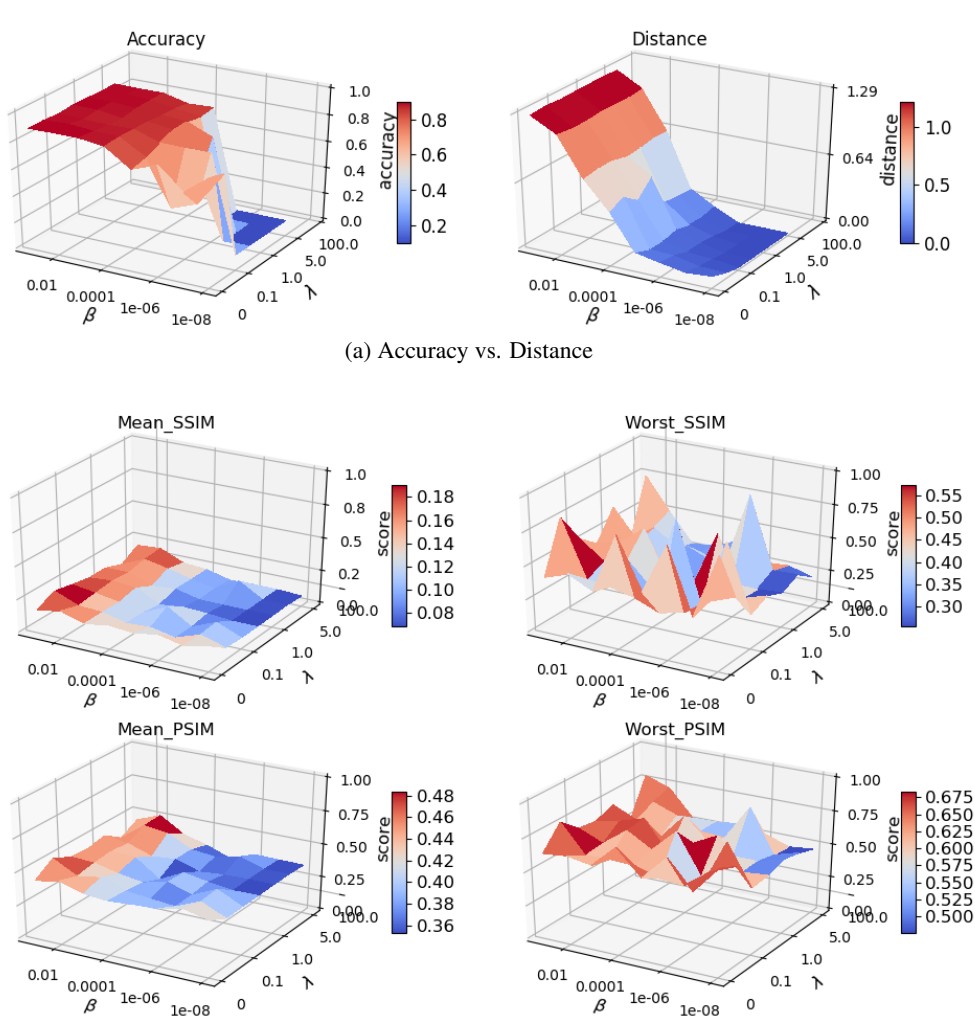

(a) Accuracy vs. Distance

(b) Quantitative evaluations on data recovery

Figure 12: Adding MixCon to the 3rd layer of CNN on FashionMNIST dataset. (a) The trade-off between data separability and data utility . We show testing accuracy and mean pairwise distance (data separability) with different $\lambda$ and $\beta$. $\lambda$ and $\beta$ show complementary effort on adjusting data separability. (b) Quantitative evaluation of data recovery results. We show SSIM and PSIM scores with different $\lambda$ and $\beta$.

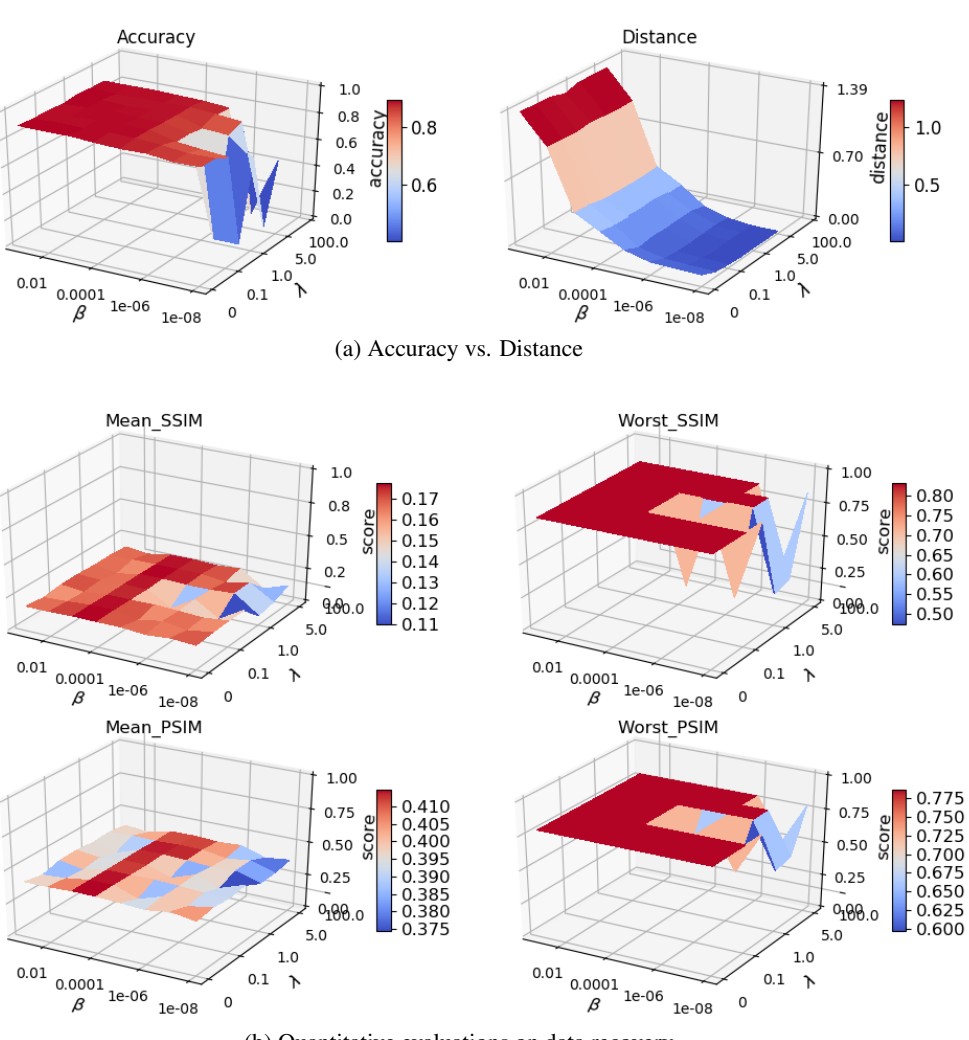

(a) Accuracy vs. Distance

(b) Quantitative evaluations on data recovery

Figure 13: Adding MixCon to the 4th layer of CNN on FashionMNIST dataset. (a) The trade-off between data separability and data utility . We show testing accuracy and mean pairwise distance (data separability) with different $\lambda$ and $\beta$. $\lambda$ and $\beta$ show complementary effort on adjusting data separability. (b) Quantitative evaluation of data recovery results. We show SSIM and PSIM scores with different $\lambda$ and $\beta$.

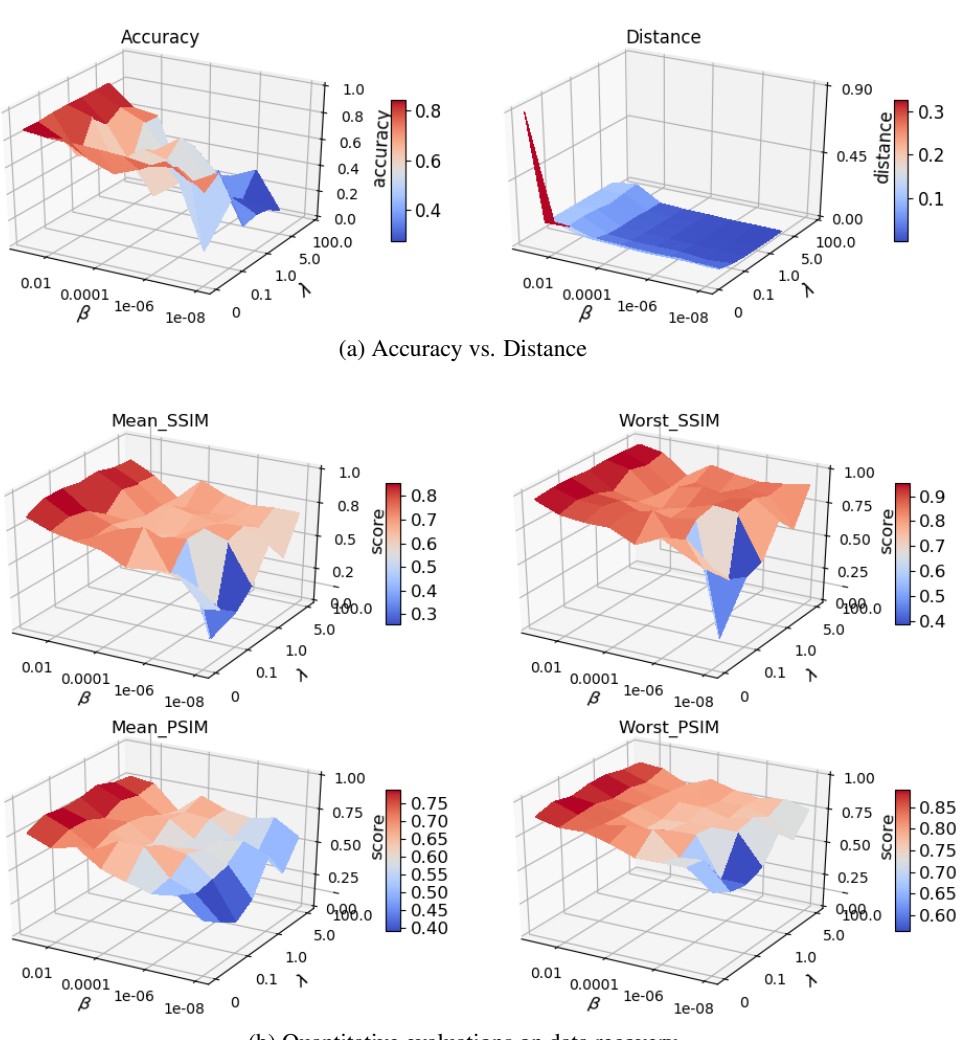

(a) Accuracy vs. Distance

(b) Quantitative evaluations on data recovery

Figure 14: Adding MixCon to the 1st layer of CNN on SVHN dataset. (a) The trade-off between data separability and data utility . We show testing accuracy and mean pairwise distance (data separability) with different $\lambda$ and $\beta$. $\lambda$ and $\beta$ show complementary effort on adjusting data separability. (b) Quantitative evaluation of data recovery results. We show SSIM and PSIM scores with different $\lambda$ and $\beta$.

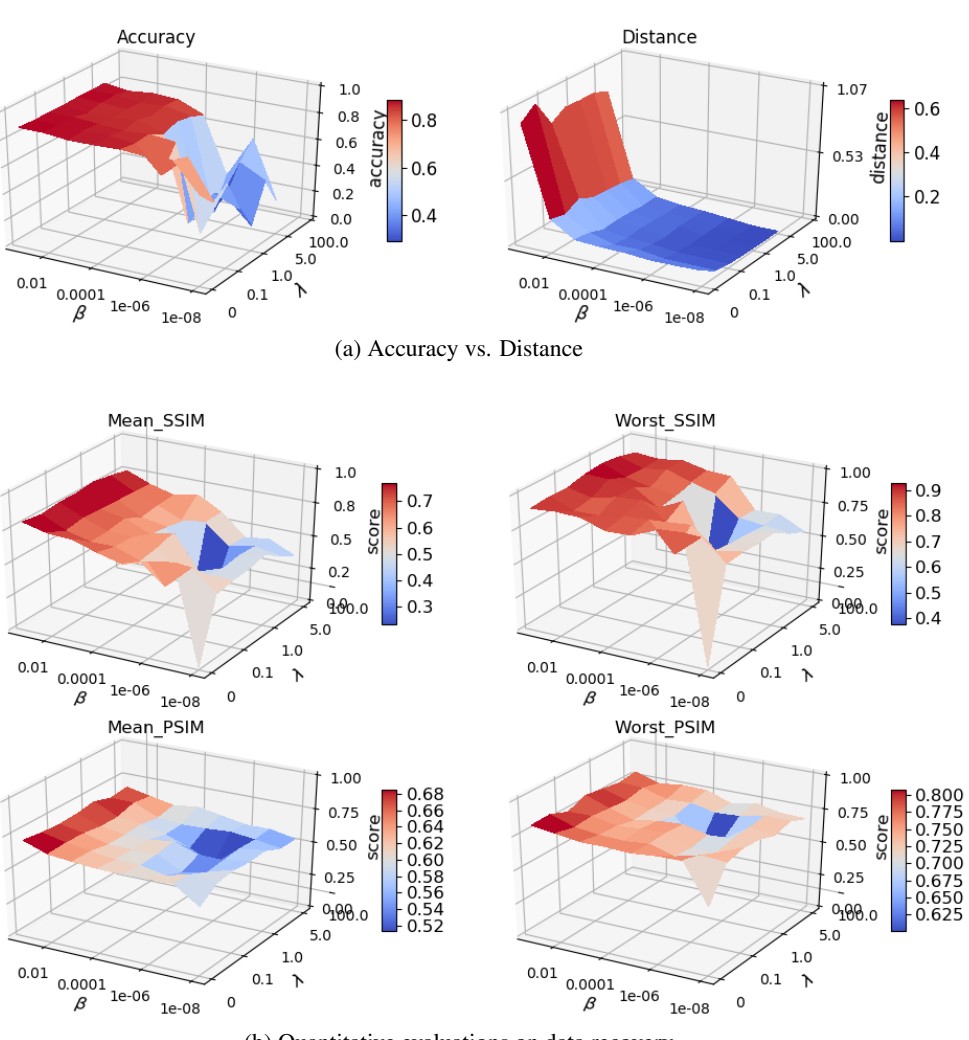

(a) Accuracy vs. Distance

(b) Quantitative evaluations on data recovery

Figure 15: Adding MixCon to the 2nd layer of CNN on SVHN dataset. (a) The trade-off between data separability and data utility . We show testing accuracy and mean pairwise distance (data separability) with different $\lambda$ and $\beta$. $\lambda$ and $\beta$ show complementary effort on adjusting data separability. (b) Quantitative evaluation of data recovery results. We show SSIM and PSIM scores with different $\lambda$ and $\beta$.

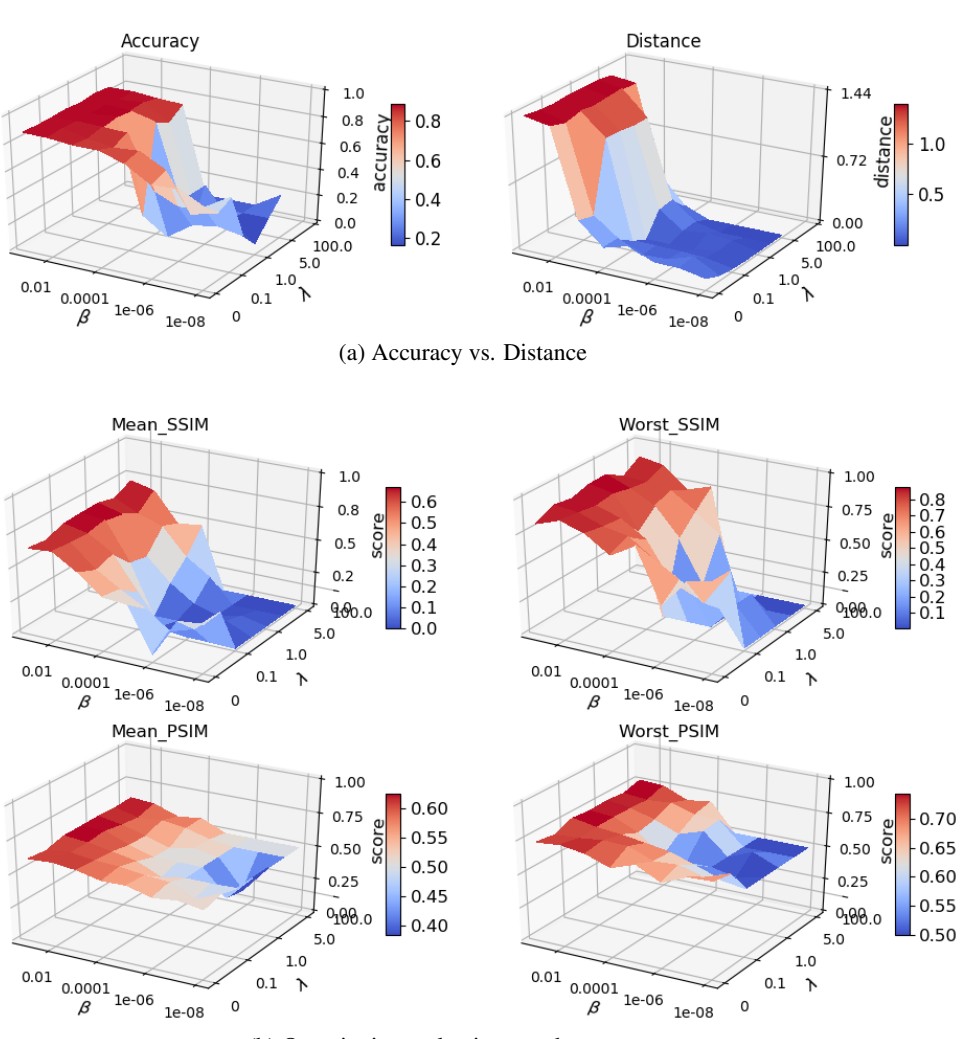

(a) Accuracy vs. Distance

(b) Quantitative evaluations on data recovery

Figure 16: Adding MixCon to the 3rd layer of CNN on SVHN dataset. (a) The trade-off between data separability and data utility . We show testing accuracy and mean pairwise distance (data separability) with different $\lambda$ and $\beta$. $\lambda$ and $\beta$ show complementary effort on adjusting data separability. (b) Quantitative evaluation of data recovery results. We show SSIM and PSIM scores with different $\lambda$ and $\beta$.

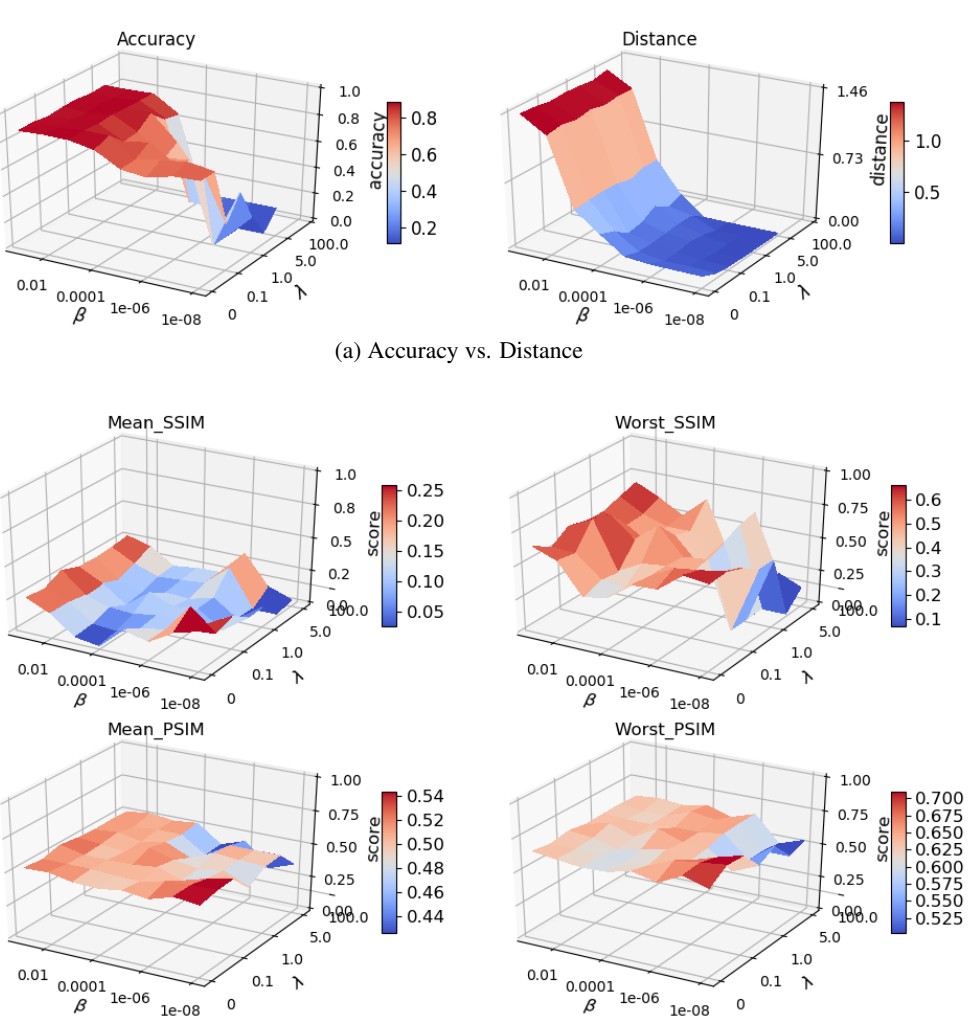

(a) Accuracy vs. Distance

(b) Quantitative evaluations on data recovery

Figure 17: Adding MixCon to the 4th layer of CNN on SVHN dataset. (a) The trade-off between data separability and data utility . We show testing accuracy and mean pairwise distance (data separability) with different $\lambda$ and $\beta$. $\lambda$ and $\beta$ show complementary effort on adjusting data separability. (b) Quantitative evaluation of data recovery results. We show SSIM and PSIM scores with different $\lambda$ and $\beta$.

