# OpenReview forum: "MixCon: Adjusting the Separability of Data Representations for Harder Data Recovery"
_ICLR.cc/2021/Conference — Reject_

### Official Review · AnonReviewer1 · 2020-10-27
**ICLR 2020 Conference Paper995 AnonReviewer**

**Rating:** 5
**Confidence:** 4

**Review:**

Post response update

Thanks for the response. However, my major concern is still that the technical contribution of this paper is limited.

-----------------------------------------------------------------------------------------------------------------------------------------------------------------------------

This paper studies an important problem, i.e., the vulnerability of deep neural networks to model inversion attacks. To address this problem, the authors design an objective function to adjust the separability of the hidden data representations as a way to control the trade-off between data utility and vulnerability to inversion attacks. Specifically, the authors propose a consistency loss (MixCon) to train data feature extractor to help protect original data to be inverted by attacker during inference. The authors also conduct experiments on synthetic and benchmark datasets to evaluate the performance of the proposed method. However, I still have the following concerns.

[-] The proposed consistency penalties (i.e., Eq. (1)) are quite straightforward and intuitional. Additionally, it lacks theoretical analysis of the proposed loss terms. For example, it would be better if the authors could analyze how much the attackers’ attacking ability are reduced by using the proposed loss terms and the approximation error before and after incorporating the consistency penalties.

[-] For the threat model, the authors assume that the attacker not only has access to extracted features but also all network parameters of the trained model. However, these assumptions are strong and unrealistic in the real-world applications.

[-] More details are needed. Firstly, in Theorem 3.2, the authors claim that under certain cases, (stochastic) gradient descent algorithm can find the global minimum of neural network function $f$. It would be better if the authors provide the proof for this theorem. Additionally, the authors mention that lower bound and upper bound imply better accuracy and hardness of inversion, respectively. However, the definitions of both lower bound and upper bound are not clearly. It would be better if the authors provide the formal (mathematical) definitions.

[-] The authors fail to cite existing state-of-the-art works on defending model inversion attacks, e.g., [1,2,3,4]. Additionally, in experiments, state-of-the-art baselines are not adopted. Currently, there are some existing defenses against model inversion attacks, e.g., [1,2,3,4]. It would be better if the authors give further discussion and compare the performance of the proposed method with that of these existing works.

[-] Some typos should be corrected. Just list some of them.
* “by proposing and a self-supervised learning based feature extractor” in Page 1.
* “force for the data representation” in Page 4.


[1] “Improving Robustness to Model Inversion Attacks via Mutual Information Regularization”, 2020.

[2] “Defending Model Inversion and Membership Inference Attacks via Prediction Purification”, 2020.

[3] “Model Inversion Attacks that Exploit Confidence Information and Basic Countermeasures”, 2015.

[4] “Adversarial neural network inversion via auxiliary knowledge alignment”, 2019.

---

> ### Author Response · Authors · 2020-11-19
> **Response to Reviewer 1**
>
> We thank Reviewer 1 for the valuable feedback and detailed summary of our paper. We have addressed all your questions in the following.
>
> > Theory for the proposed loss term
>
> Thanks for recognizing the effectiveness of our proposed penalty term. The theoretical analysis was provided in Appendix B (Theorem B.10) of our original submission. We showed that if the distance is sufficiently small, then the recovery problem is computationally hard and any recovery algorithm would run in exponential time in the worst-case scenario. In particular, if we take the lambda term to be sufficiently large, then we expect in the global optimum, the pairwise distance is small.
>
> It is very challenging to explicitly quantify the attackers’ attacking ability, due to the variety of attacks and especially different threat models they assume (e.g. some attacks may assume prior knowledge or side-channel information, which cannot be quantified by a theoretical formulation). We believe providing a unified theoretical framework would be helpful, but it is beyond the scope of this work, which aims to study whether there is an effective way to adjust data separability for harder inversion under a clearly defined threat model. A common way to define it is to use computation hardness as evidence, like our Theorem B.10.
>
> >  Assumption of the attack model
>
> We agree that the assumptions are strong as we intentionally choose the strongest possible attack to show the upper bound of attack performance in any possible real-world scenarios. Also, such white-box settings have been used in [5,6] that focus on sample-wise data recovery.
>
> > Clarification on Theorem
>
> Thanks for pointing it out. Theorem 3.2 is credited to Allen-Zhu (2019b) [7], we are sorry to inadvertently omit the citations and have placed it back in our revision. The proof can be found in [7]. We use it here for intuition and explain why data separability is considered to be important for both successful training and defending model inversion attack. The definition of the lower bound and upper bound can be referred to Definition 3.1 of our original submission. We have clarified the definitions in the updated manuscript.
>
> > Cite and compare SOTA inversion defending methods
>
> Thanks for pointing to the works. [3] was indeed cited in our submission. [4] proposes an interesting black-box inversion attack method. But [3-4] are attack papers that may not be suitable for comparison. [1-2] are interesting defense papers (though in preprints). We have included them as related work in the updated version. However, there are fundamental differences in the problem setting between [1-2] and ours, which makes an empirical comparison unfair, specifically:
>
> - Our MixCon loss is proposed to be added to the middle layers. But the adversary in [1-2] only has access to the output of the final layer.
>
> - [1] is a concurrent work (it went on arxiv about two weeks before the submission deadline). It is great to know this work, but this paper has not provided source code. A fair comparison will be left as future work.
>
> - In [2], the defense either requires an argmax operation on output or needs knowing attack function, which can not be directly applied to the middle layer or compared with our simple setting.
>
> We thank Reviewer 1 again for the valuable comments and careful review. We have corrected the typos and will thoroughly proof-read the paper in the final version.
>
> ------
> [1] Zhang et al. 2020. Improving Robustness to Model Inversion Attacks via Mutual Information Regularization
>
> [2] Zhang et al. 2020. Defending Model Inversion and Membership Inference Attacks via Prediction Purification
>
> [3] Fredrikson et al. 2015. Model Inversion Attacks that Exploit Confidence Information and Basic Countermeasures
>
> [4] Zhang et al. 2019 Adversarial neural network inversion via auxiliary knowledge alignment
>
> [5] He et al. 2019, Model inversion attacks against collaborative inference
>
> [6] Zhang et al. 2020, The secret revealer: generative model-inversion attacks against deep neural networks
>
> [7] Zhu et al. 2019, On the convergence rate of training recurrent neural networks

---

### Official Review · AnonReviewer2 · 2020-10-27
**interesting problem of inversion attack is studied with not a convincing approach**

**Rating:** 5
**Confidence:** 3

**Review:**

The paper aims at strengthening DNN against inversion attack. It proposes to utilize an extra term in NN training objective function, called L_mixcon, to play with separability of  hidden representation of data in different classes. Following is my concern:
-	Reducing hidden representation separability of data points in different classes equates to more confusion for classification. All the results shown in the paper, e.g. Table 3 or Figure 4, confirm that and show that there is a trade off between accuracy and robustness against the inversion attack. Thus, this proposed method could be helpful if the user is willing to give up on some accuracy in the hope of getting a more robust model. However, the paper does not highlight this fact and presents the method as if it provides the same accuracy with higher robustness. For example in page 8 it is stated that : “We select (lambda, beta) to match the accuracy results of MixCon to be as good as Vanilla training (see Accuracy in Table 3),” whereas in Table 3 the accuracy is not as good as Vanilla, and it is misleading.
-	It is also obvious from formula (1) that minimizing L_mixcon with lower beta or higher lambda degrades the accuracy, and the provided results show the same thing. But it is not shown that the sweet spot exists as there is always a trade-off.
-	Regarding formula 1:
o	Why do we only look at p data points per class and not all?
o	The formula says i-th data point form class c_1 would be compared against i_th data point from class c_2. How do you do this one-to-one mapping between points of different classes? Or did you mean all pairs and the formula is not written properly?
o	Also, as a suggestion, I think designing beta as a function of classe (c) would be more appropriate.
-	Regarding the layer h, I am wondering if there is any recommendation how to choose that and how many layers to choose. In this paper only one layer is considered. I can also imagine the best choice of beta and lambda depends on the choice of layer.
-	It is mentioned that the local feature extractor is a shallow NN in the setting, is it one layer CNN with nonlinearity or a linear model?
-	The paper is understandable but there are multiple typos and the English could be improved.

---

> ### Author Response · Authors · 2020-11-19
> **Response to Reviewer 2**
>
> We thank Reviewer 2 for the valuable feedback. We have addressed all your questions in the following.
>
> > Claim about the same accuracy or scarifying some accuracy
>
> Thanks for your valuable suggestions. We have addressed that “our proposed method is helpful if the user is willing to give up on some accuracy in the hope of getting a more robust model”  in our revision.
> Nevertheless, the accuracy drop can be negligible in some experiments. For the results reporting in Table 3, the accuracy of Vanilla vs. our MixCon for the three datasets are: MINIST-- 99.1% vs. 98.6%, FashionMNIST -- 89.8% vs. 88.9%,  and SVHN 88.4% vs. 88.2%. The accuracy drop of using MixCon is within 1.0%, which is acceptable for most of the applications.
>
> > Definition of sweet-spot
>
> We agree that there is a trade-off, but having a trade-off does not mean not existing sweet spots. The trade-off between the accuracy of defending inversion attack shows a nonlinear relationship that can be inferred from Fig.3 (main text) and Fig.6-17 (Appendix D2). The sweet spot is in the space where the accuracy degradation curve is slow, while recovery computational complexity increases fast (i.e., reflected by a small distance and small similarity scores).
>
> In the updated version, we have clarified the definition of the sweet spot. By sweet spot, we mean that the set of (beta, lambda) that suffers from negligible accuracy loss (say within 1%) and the model inversion becomes significantly harder w.r.t computational complexity or breaks the attack (less similarity to the original input data).
>
> > Investigation on different cut layers
>
> Thanks for the question. First, in a real-world distributed learning scenario (collaborative inference), the “cut layer” is pre-defined by the participants (clients and the server). Second, there are no advantages of adding MixCon to the many layers before the “cut layer” to defend against inversion attacks, because the attacker is not able to get access to the original data hidden representations from those layers.
>
> We agree with R2 that the best (lambda, beta) for different layers might be different, which may require hyperparameter selection in real-world applications.  Although those are not the focus of the work, showing the results will be valuable. We have added the results in Appendix D.2 of our updated version. Usually, the layer further away from the input is harder to invert. Given the accuracy drop tolerance is 1%, here we list the best (beta, lambda) on MNIST that result in the smallest mean similarity scores to reflect the overall trend for each layer.
>
> | Layer                |      1      |      2     |      3     |      4      |
> |----------------------|:-----------:|:----------:|:----------:|:-----------:|
> | ($\lambda$, $\beta$) | (1e-1,1e-3) | (1， 1e-4) | (2， 1e-2) | (1e2， 1e-6) |
> | Acc                  |    98.79    |    98.3    |    98.1    |     98.3    |
> | SSIM                 |     0.77    |    0.15    |    0.08    |     0.08    |
> | PSIM                 |     0.95    |    0.52    |    0.36    |     0.51    |
>
> > Clarification on feature extractor
>
> In our synthetic experiment, the local feature extractor is a two-layer fully connected network. In our benchmark experiment, we used standard NN and the local feature extractor is a two-layer convolutional network. They are both nonlinear.
>
> > Clarification on formula 1
>
> p is the data size for the smallest class (in practice, it is the smallest class size in a minibatch when using SGD optimizer), and we truncate the large class(es) for ease of calculation. Our Eq1. exactly reflects all pairwise distance through one-to-one mapping. Each mini-batch contains an equal number of samples of each class. We order the data points into data arrays for each class, i.e. if we have class {a b c} and each mini-batch contains 12 data points (4 data points per class), then we order the data points as [[a1,b1,c1],[a2,b2,c2],[a3,b3,c3],[a4,b4,c4]]. In each small [ai,bi,ci], we perform pairwise comparison (easy to implement using tensor broadcast operations). Thus we do the one-to-one mapping between the training points of different classes in a training step. In the training process, we randomly shuffle the data. Due to the random shuffling, the mappings are not fixed for the whole training process.
>
> Thanks again for your careful review. We will correct the typos and thoroughly proof-read the paper in the final version.

---

> > ### Comment · AnonReviewer2 · 2020-11-20
> > **After authors' responses**
> >
> > Thanks for the response.
> > - regarding formula 1: then you need to clarify that in the text and the formula should reflect that. Also, truncating the dataset brings up the question on how much information you would lose and what is the subsequent effect?

---

> > > ### Author Response · Authors · 2020-11-21
> > > **Thanks for your reply and suggestion**
> > >
> > > We appreciate Review 2's prompt reply.
> > >
> > > Following your suggestion, we have addressed shuffling data under Eq.1 and added the details about mini-batch training in the experiment in our updated version.
> > >
> > > Sorry for the confusion. Truncation in a mini-batch does not mean throwing away any data for the whole training process. In our benchmark experiment, the data of different labels are almost uniformly distributed, and the number of omitted data points for computing MixCon loss in one batch is small. Also, following the regular training routine, data is shuffled in each iteration, and the model is trained for many epochs. We argue that all the data points can be visited during the whole training process. Thus the information loss can be negligible. For the unbalanced situation, we can repeat sampling the same data in the small class(es).
> > >
> > > Thanks again for your valuable feedback.

---

### Official Review · AnonReviewer4 · 2020-10-31

**Rating:** 5
**Confidence:** 3

**Review:**

Post response update

I would like to thank the authors for their detailed response. The response addresses my confusion around the use of the terminology of model inversion (I would further suggest that the author use the term data reconstruction rather than than model inversion to avoid readers misunderstanding model inversion as referring to [1]). I still have concerns around the fact that differential privacy is not used as a baseline here, which would strengthen the argument made in the response that it provides orthogonal guarantees.

-----------

The paper looks at the tradeoff between data separability in an embedding space and vulnerability to model inversion. The paper hypothesizes that increasing data separability improves accuracy but exposes the model to model inversion attacks.

- At a high level it is not clear why the model inversion is a well-motivated attack vector given that model inversion extracts an average representation of the points from a class, not specific training points. The introduction talks about data recovery but it is not clear what it means to “recover” a data point in the context of this work.
- Grammar in page 1: “ The central question here is how to better protect the data from being reconstruct while keeping useful information to the classification task.”
- Grammar in page 1: “ by proposing and a self-supervised learning- based feature extractor”
- Unclear what the following means: “  At a high level, we focus on the pipelines combining local data representation learning with global model learning manner.”
- Section 3.1 proposes data separability as a measure of privacy, but it is not clear why it is necessary to introduce a new definition for privacy. Why not consider well-established definitions such as differential privacy?
- Section 3.1 proposes informal statements tying data separability but the statements are not demonstrated, and the analysis does not outline how the results would be proved (There is also not proof in supplementary materials).
- Section 3.2 introduces concepts such as “confusion” in embedding space without defining them. Moving forward, making the claims more precise would help make the paper more readable.
- How does the approach from Section 3.2 relate to other losses like the triplet loss which compare distances between different points in the embedding space?
- Section 4.1 does not specify whether the input x is part of the training and test set. Is the goal of the model inversion to invert a training point or instead a test point which the model is inferring on?
- It is not clear why the experimental setup from Section 4.2 is well-motivated to study the model inversion problem given that the model architecture being fully connected, and the dataset being synthesized, but not used in prior work.
- In Section 4.2, it is unclear how the quality of a reconstruction is evaluated. Is a successful model inversion evaluated based on human perception and a similarity metric? Does the lack of similarity mean that there is no privacy leakage? Here it seems that this comes back to the definition of privacy used, which looks at average case rather than worst case.
- The introduction discussed a distributed setting scenario but this does not seem to be considered in the experimental setup itself.

---

> ### Author Response · Authors · 2020-11-19
> **Response to Reviewer 4 -- [1/2]**
>
> We thank reviewer 4 for the valuable comments. We would like to clarify that the focus of our paper is on defending sample-specific model inversion attacks (a.k.a data recovery) but not to guarantee (differential) privacy. Model inversion attack is well motivated and studied in [1,2,3]. Following your suggestion, we have emphasized and clarified our focus in the updated version.
>
> > The motivation of inversion attack
>
> Sorry for the confusion. “Given that model inversion extracts an average representation of the points from a class” might be a misinterpretation of the model inversion attack setting in our paper, which is a **sample-specific** one following [2].
> We specified the inversion attack in Section 4.1, the model inversion attack works in the following way: given h(x) of an x, and parameters of h, where h is the hidden-layer representation function, the attacker aims to recover x’ in an approximation of x. This means the attacker aims to recover individual data points instead of extracting the model or an average representation of the points from a class. Also, our theory analysis serves for single data inversion. The motivation of inversion attack is to evaluate the recovery of single points. We have clarified in the introduction of our updated version that “We use the model inversion attack that reconstructs individual data”.
>
> >  Definition of privacy and quality evaluation in Sec 4.2
>
> We do not aim for a new definition of privacy. Instead, we focus on defending a form of model inversion attacks -- data recovery, but not general privacy. There are substantial differences between differential privacy and model inversion, i.e., differential privacy cares about a single sample's effect on the aggregate result. In contrast, model inversion attack focused on our work target on a given sample without considering population. We agree with R4 that the word "privacy", which was used once in our original submission might be confusing. We have rephrased the sentence as "The question is, what is the right measure for the amount of information of successful classification and defending against data recovery?"
>
> We agree with you that the lack of similarity does not mean that there is no privacy leakage. However, our main message is to adjust data separability for defending sample-specific inversion attacks. Thus our experiments use similarity to evaluate instance-wise data recoveries, but not for quantifying privacy leakage.
>
> For our benchmark experiment, the comprehensive evaluation is based on human perception and similarity metrics to measure the similarity of instance-wise inversion results and their original input. A similarity measurement on single data reveals the ability to defend against inversion attacks, as they have been widely used in [2,4,5].
>
> “The definition of privacy used, which looks at average case rather than worst-case” might be a misinterpreting for our attacker setting. In fact, our attack experiment is on single data points, and it can look at both average case and worst case, as we reported in Table 3.
>
> > Proof of Theorem 3.1
>
> Sorry for the confusion. Theorem 3.1 is credited to [6], and we overlooked the reference in our original submission. We have added the reference in the updated version. The proof can be found in the original paper [6].
>
> > Clarification on “confusion” and input x
>
> Thanks for the questions. “Confusion” means that the embedded features of different classes are getting closer to each other in the features space. We provided the inversion results compared with minimizing distance within class only in appendix D.1, showing MixCon makes data harder to invert.
>
> The input x is a test sample.  The goal of the model inversion is for a test point. In Section 1 and Section 3, we stated “We focus on defending data recovery during inference.” We have clarified it in our updated version.
>
> -------
> [1] Fredrikson, et al. 2015, Model inversion attacks that exploit confidence information and basic countermeasures.
>
> [2] He et al. 2019, Model inversion attacks against collaborative inference.
>
> [3] Zhang et al. 2020, The secret revealer: generative model-inversion attacks against deep neural networks
>
> [4] Park et al. 2019, An Attack-Based Evaluation Method for Differentially Private Learning Against Model Inversion Attack
>
> [5] Wu et al. 2020, Evaluation of Inference Attack Models for Deep Learning on Medical Data
>
> [6] Zhu et al. 2019, On the convergence rate of training recurrent neural networks

---

> > ### Author Response · Authors · 2020-11-19
> > **Response to Reviewer 4 -- [2/2]**
> >
> > > Difference between MixCon and other losses, like Triplet loss.
> >
> > Triplet loss [7] aims to minimize the distance between the positive samples and anchors while maximizing the distance between the negative samples and anchors. Different from Triplet loss, we do not have the concept anchor and do not define positive and negative samples.
> > Also, different from contrastive loss [8], we try to minizine the distance of data pairs (X1, X2), where they come from different classes instead of from the same class in [8]. To ensure the model is still trainable, we ensure the minimal distance of the data points from different classes should be larger than a certain margin, which is given Theorem 3.2.
> >
> > > Motivation of simulation experiment.
> >
> > The hypothesis of this paper is that adjusting separability has an impact on the ability of model inversion attack.  We present the synthetic data experiment because of the following insights:
> >
> > - The evaluation of inversion attack is not the main purpose of the synthetic setting, but we show it for completeness of our evaluation.
> > - The main purpose is to show an illustration (or visualization) of how the MixCon affects the embedding space, with a hidden layer of two neurons. We believe this visualization is important for understanding the relationship between separability, invertibility and utility. However, we found this visualization to be NOT straightforward with real-data, which usually requires complex neural network architecture and wide hidden layers.
> > - We intentionally choose the low-dimensional hidden-layer and the simple fully connected network to allow precise manipulation of separation for better visualization.
> >
> > We have clarified the points above in the updated version.
> >
> > > Distributed learning setting in the experiments
> >
> > As we stated in the introduction “service provider trains and splits a neural network at a “cut layer”, then deploys the first to cut layers to clients. Clients encoder their dataset using those layers, then send the data representation back to the cloud server using the rest of the layers for inference [9,10,11]”.
> > The focus of this paper is not distributed learning itself, but the data recovery issue under the setting. Without loss of generality,  our experiments run with one client and one server (exactly as we split a network f(x) into sever--g(h) and client--h(x) two parts), which is the simplest setting to answer the evaluation questions.
> >
> > > Typos and other remarks
> >
> > Thanks for your careful review. We will correct the typos and thoroughly proof-read the paper in the final version.
> >
> > ------
> > [7] Schroff et al. 2015, FaceNet: A Unified Embedding for Face Recognition and Clustering
> >
> > [8] Hadsell et al. 2006, Dimensionality Reduction by Learning an Invariant Mapping
> >
> > [9] Teeraputtayanon et al. 2017, Distributed deep neural networks over the cloud, the edge and end devices
> >
> > [10] Ko et al. 2018, Edge-host partitioning of deep neural networks with feature space encoding for resource-constrained internet-of-things platforms
> >
> > [11] Vepakomma et al. 2018, Split learning for health: Distributed deep learning without sharing raw patient data

---

### Author Response · Authors · 2020-11-19
**Thanks for your valuable feedback**

We would like to thank all the reviewers for taking the time to contribute their insightful comments, which helped us improve the paper. We also appreciate all the reviewers recognizing that the problem we are tackling is important and the method we proposed is intuitive. We are delighted to highlight our theoretical contributions to understand why our MixCon penalty that adjusts data representation separability can work for defending inversion attacks. Our detailed point-to-point responses can be found below, and we have also carefully updated the manuscript to follow the constructive suggestions from the reviewers.

---

### Decision · Program_Chairs · 2021-01-07
**Final Decision**

**Decision:**

Reject

**Comment:**

In this paper, the authors change the loss function of NNs to reduce the separability of the different classes in one of the hidden layers. The rationale for this assumption that the trained network will be more robust against white-box model inversion attack. The reviewers all concur that the paper had some merit, but that the paper is not well presented and believe the paper is not ready to be presented at ICLR.

Also, the separability issue is not totally explained, because a reduced L2 norm might not be the whole story that explains why a white-box model inversion would rely on for leaking information. This might need to be proof further and a couple of experiments in which there is still leakage of information shows the additional robustness from the new penalty.